# Rapid retreat of Berry Glacier, West Antarctica, linked to seawater intrusions revealed by radar interferometry

Hanning Chen [1] ✉, Eric Rignot [1,2,3] ✉, Bernd Scheuchl [1], Pietro Milillo [4], Mathieu Morlighem [5], Ratnakar Gadi[1], Enrico Ciracì [1,2], Jae Hun Kim[1] & Luigi Dini [6]

We employ a time series of ERS-1/2, ALOS-1/2 PALSAR, Sentinel-1, COSMO-SkyMed, and RCM differential synthetic-aperture radar interferometry data from 1996 to 2023 to document the short and long-term migrations of the grounding line (GL) of Berry Glacier, West Antarctica, a tributary of Getz Ice Shelf that controls 10% of its ice discharge. In 2019–2021, we detected a short-term GL migration of $18.0 \pm 0.9$ km, which is exceptionally long and implies that the glacier bed is up to 1300 m deeper than previously known. On short time scales, the GL migrates between three states controlled by bed topography. The observed flexing of the glacier suggests that seawater is trapped in the newly formed ice shelf cavity in an irregular fashion during the tidal cycle. From 1996 to 2021, the most inland position of the GL retreated by $18.1 \pm 0.4$ km, or 0.7 km/year, the ice thickness decreased by $11 \pm 1$ m/year, the ice sped up by $64 \pm 5\%$, and the glacier lost a total mass of $131 \pm 23$ Gt. We attribute the rapid retreat to an enhanced ocean heat flux from warm Circumpolar Deep Water (CDW) reaching the grounding line through favorable bathymetry channels, combined with km-sized seawater intrusions beneath the glacier that cause rapid melting of basal ice.

The Antarctic Ice Sheet has been a major contributor to sea level rise in the last four decades[1], but projections of its evolution in the coming century are affected by large uncertainties[2]. Central to these uncertainties is the modeling of the evolution of the grounding line (GL), i.e., the transition boundary between grounded and floating ice. The ice that crosses the GL floats in the ocean waters. Changes in GL position affect the basal friction regime of the glacier, which, in turn, controls its speed, ice discharge, and mass loss[3]. During a GL retreat, the basal resistance may drop and the glacier may accelerate. In contrast, during an advance of the GL, resistance to flow may increase, and the glacier may slow down. Most numerical ice sheet models do not account for any short-term variability in GL position (e.g., due to ocean tides) and

should not apply basal melt at a fixed GL[4]. In this study, we revisit this assumption with novel interferometric synthetic aperture radar (InSAR) observations.

Global Positioning System (GPS), tilt meters, satellite altimetry[5], and InSAR data[6] have been used to delineate GL positions. Altimetry detects vertical motion below a satellite track with a precision of 5–10 cm[7]. InSAR detects vertical motion on a 100 km wide strip with a precision of 2–3 mm[8]. InSAR GL mapping started in 1995-1996 with Earth Remote Sensing Satellites 1 and 2 (ERS-1/2) in a 1-day repeat cycle and resumed in 2014 with Sentinel-1a (S1a) in a 12-day repeat cycle[9] and Sentinel-1b (S1b) in 2016, in a 6-day repeat cycle until the loss of S1b in December 2021. Here, we also include data from the ALOS-1/2 PALSAR

[1]Dept. Earth System Science, University of California, Irvine, Irvine, CA, USA. [2]Jet Propulsion Laboratory, California Institute of Technology, Pasadena, CA, USA. [3]Dept. Civil and Environmental Engineering, University of California, Irvine, Irvine, CA, USA. [4]Cullen College of Engineering, University of Houston, Houston, TX, USA. [5]Dept. Earth Sciences, Dartmouth College, Hanover, NH, USA. [6]Space Geodesy Centre Giuseppe Colombo, Italian Space Agency, Matera, Italy. ✉e-mail: hanninc@uci.edu; erignot@uci.edu

mission for the years 2008–2009 and 2021, a time series from the COSMO-SkyMed (CSK) constellation at a 1-day repeat cycle for the years 2020–2021, and data from the RADARSAT Constellation Mission (RCM) in a 4-day repeat cycle for the year 2023. We characterize the short- and long-term variabilities in the GL position of Berry Glacier, which is a part of the Getz Ice Shelf, West Antarctica. Short-term variability is used to define an "Ice Grounding Zone" (IGZ), or a region of tidal-scale variations in the position of the GL.

Getz drains an area of 85,900 km² that contains an ice volume equivalent to a 22-cm rise in sea level. Berry Glacier drains a sector 5593 km² in size into the Getz Ice Shelf and controls 10% of the total ice flux (Fig. 1, Fig. 2). Berry flows at 760 m/year in the IGZ and holds an ice volume equivalent to a 1.5-cm rise in sea level. Near the GL of Getz, many glaciers are in contact with warm waters coming from Circumpolar Deep Water (CDW), which rapidly melt ice from below[10–13]. As a result, the Getz Ice Shelf is one of the largest producers of basal meltwater in Antarctica, which contributes to weakening ice shelf buttressing and enhancing dynamic mass loss[14].

## Results and Discussion
### Grounding line migration
Using a time series of InSAR data, we find that the length of the IGZ increased from 0.6 km in 1995/1996 to 18.0 ± 0.9 km in 2019–2021 (Fig. 1). Because we have only two interferograms in 1995/1996, the length of the IGZ is defined with less precision, but we find no residual signal upstream of state 1 in 1996 that would suggest that seawater could have intruded even farther upstream back in the 1990s (Supplementary Fig. 1a, b). In contrast, the most recent interferograms always display residual vertical motion between states 1 and 3 of the GL. We conclude that seawater intrusions were unlikely upstream of

state 1 in 1996 (Fig. 3). In 2019–2021, the IGZ is considerably longer and the longest recorded in Antarctica[15–17]. Calculations of ice thickness assuming hydrostatic equilibrium[18] using a WorldView (WV) DEM from year 2019, however, indicate that the IGZ should only be 300–600 m long (Fig. 3a). This discrepancy imposes a revision of the Berry bed topography in BedMachine v3.7 (BMv3.7), which is not derived from mass conservation in this area but from an interpolation between NASA Operation IceBridge (OIB) profiles.

In terms of long-term migration, we find that the upper limit of the IGZ retreated 18.1 ± 0.4 km from 1996 to 2021, i.e., the lower portion of the 2021 IGZ matches the upper limit of the 1996 IGZ, while the upper limit is 18 km further inland than in 1996. During the more recent time period 2009-2021 with denser observations, the GL retreated 7.0 ± 0.4 km, or 0.5 km/year (Fig. 1) but also started to migrate laterally by about 3 km to the west into an unnamed tributary glacier (Fig. 4). Another 1.5 km retreat of the upper limit of the IGZ occurred between 2021 and 2023 (Fig. 1).

We find excellent agreement between the vertical motion of the ice surface measured with the double difference interferograms and changes in sea surface height (SSH) calculated using the CATS2008 tidal model[19] corrected for changes in atmospheric pressure, or the inverse barometer effect (IBE), calculated using the ERA5 pressure fields[20] (Fig. 3c). We find no correlation between the position of the GL and SSH (Fig. 3d), even if we take into account the spring-neap cycles and ebb/flow (Supplementary Fig. 2), i.e., the GL does not migrate in phase with SSH as we would expect in a freely opening and closing cavity. Instead, the GL oscillates between three states: (1) state 1 near the 1996 IGZ, (2) state 2, about 6 km upstream, along a retrograde portion of the new bed that is discussed later, and (3) state 3 at the southern limit of the 2021 IGZ or 2 km further inland. The three states

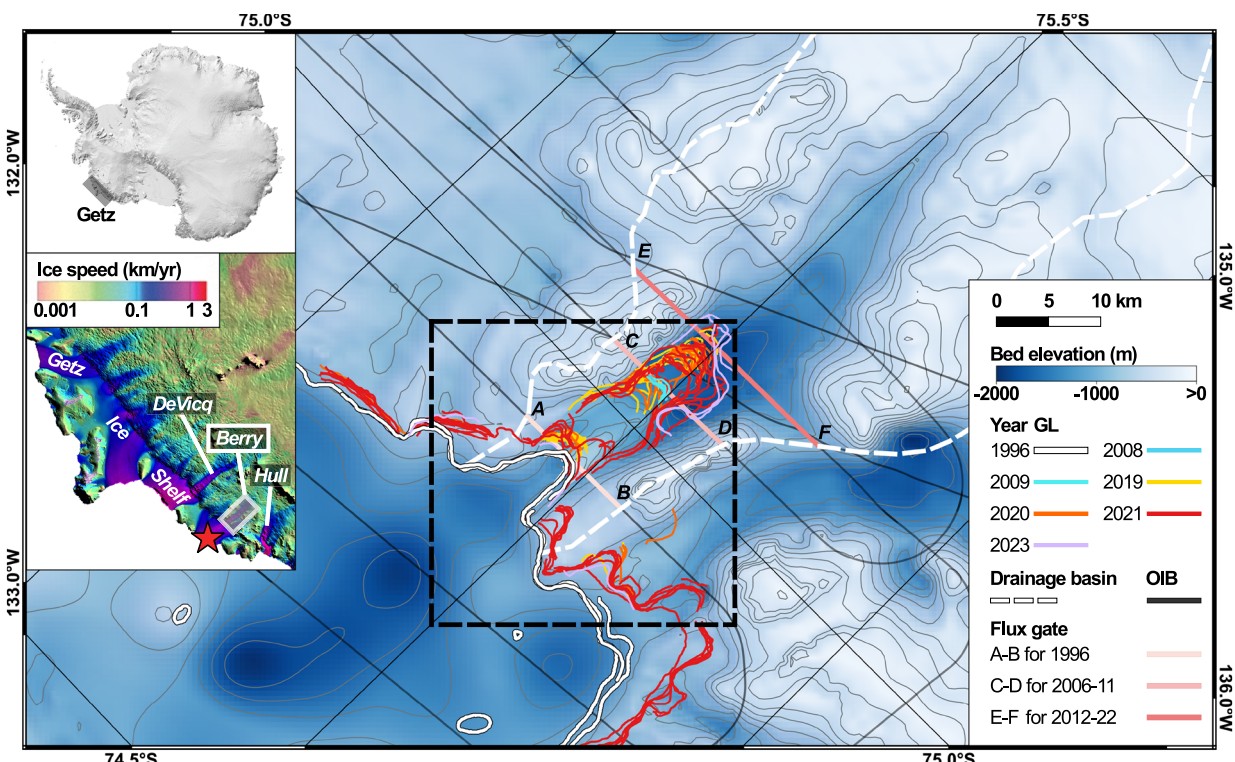

**Fig. 1 | Grounding line (GL) positions of Berry Glacier, West Antarctica.** 93 GL positions (location of Berry in inset) for years 1996 (white) from ERS-1/2, 2008 (blue)/2009 (cyan) from ALOS-1 PALSAR, 2019 (yellow) from S1a/b, 2020 (orange) from CSK, 2021 (red) from ALOS-2 PALSAR/CSK, and 2023 (purple) from RCM data overlaid on bed topography from BedMachine v3.7 (outside IGZ)[52] and deduced from flotation using a WorldView DEM (inside IGZ), color coded from -2000 m (blue) to 0 m (white), with 100-m contour levels. Red star in the inset shows where we calculate tides at the ice front[19]. NASA Operation IceBridge (OIB) tracks from 2011-2016 are black lines. Flux gates for ice discharge are pink. The drainage basin is a white dash line. The black dash box delineates area in Fig. 2a, c, e.

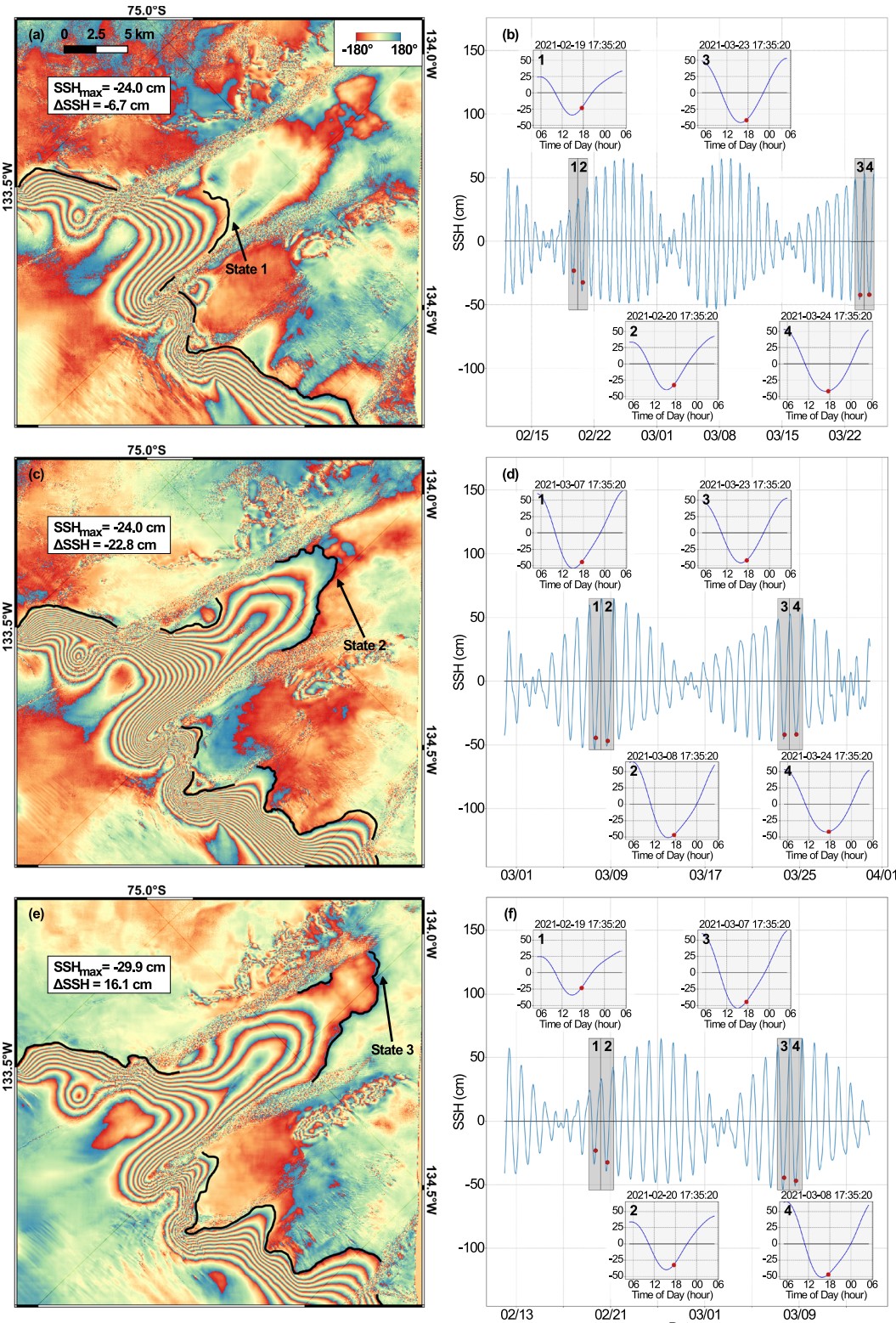

**Fig. 2 | Examples of three states of grounding line (GL) positions of Berry Glacier, West Antarctica. a**, **c**, **e** GL positions (black line) between states 1-3 overlaid on CSK interferograms combining 4 tidal states in (**b**, **d**, **f**). Color bar is the phase difference from -180 to +180 degrees; **b**, **d**, **f** SSH (red dot) at the time of acquisition of each scene used for the differential interferogram within a ±12-hour window.

stand out in the density map of GL positions retrieved for 2019–2021 (Fig. 3a). We conclude that the GL is located in state 1 in 1996, reaches state 2 around the year 2008, and is located predominantly in state 3 in 2019–2023 with possibilities for the ice to touch the seabed at state 1 and state 2 on occasions during the tidal cycle.

## Ice dynamics

We compare the GL retreat for 2006–2022 with changes in the elevation of the ice surface, $h$, the thickness of the ice, $H$, and the speed of the ice, $V$. Between states 1 and 2 along the center line, $h$ decreased by 24 m in 2006–2022, equivalent to a reduction in thickness $H$ of 222 m

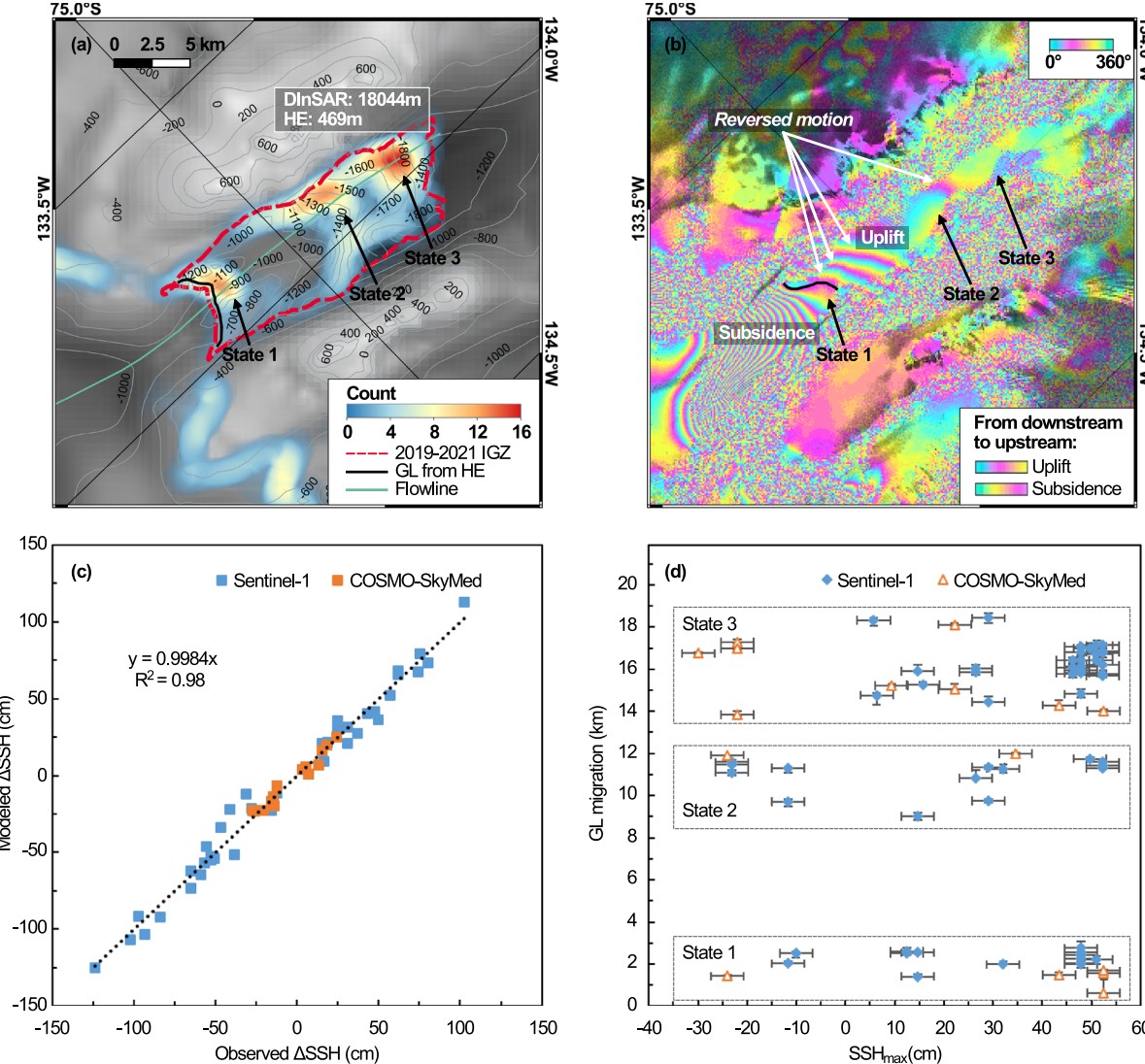

**Fig. 3 | Grounding line (GL) migration of Berry Glacier, West Antarctica. a** GL density and limits of IGZ (red dotted line) in 2019-2021, GL from hydrostatic equilibrium (HE, black) in year 2019 and center flowline (green) overlaid on bed topography with 200-m contour levels; **b** Sentinel-1 differential interferogram from year 2019 with GL position (black) and upstream phase signal in the opposite sense motion; **c** modeled differential sea surface height (SSH) versus observed InSAR displacements for S1a/b (orange) and CSK (blue); **d** GL position versus maximum SSH for states 1-3 during the 3 (to 4) acquisition times. Vertical error bars indicate the farthest and closest distances that the GL may reach based on the fringe pattern. Horizontal error bars denote uncertainty estimates derived from the CATS2008 tide model and the ERA5 reanalysis.

due to flotation, or a thickness thinning rate of 13 m/year. The lowering of the surface of the ice reaches 35 m along the west flank of Berry, where the IGZ migrated laterally, which is equivalent to a thickness reduction $dH/dt$ of 19 m/year (Fig. 4a). During 2006–2022, the glacier speed increased by 65 m/year at the center between states 1 and 2, and 136 m/year along the west flank (Fig. 4b). Glacier speed-up is, however, the largest between state 2 and state 3 and extends 28 km beyond the 2021 IGZ (Fig. 4b). In that region stretching from state 2 to 28 km beyond state 3, the speedup averages 134 m/year, with a maximum of 224 m/year between 2006–2022.

From 1996 to 2022, along the G-H profile, the surface elevation decreased by 59 m and the speed increased by 48% from 448 m/year to 665 m/year (Fig. 4c,e). Along profile C-D, the acceleration is 52%, accompanied by rapid thinning, especially on the central and western sides (Fig. 4d, f). On average in the 2021 IGZ between 1996–2022, the ice thickness dropped at a rate of 11 ± 1 m (the surface elevation changed by 47 ± 3 m) and the increase in speed is 64 ± 5%.

We calculate ice discharge in state 1 in 1996, state 2 in 2006–2011, and state 3 in 2012–2022. The gaps in the time series of thickness and speed are filled with linear interpolation. We calculate anomalies in ice discharge, that is, excess ice discharge compared to average surface mass balance for 1979–2004 that would keep the glacier in a state of mass balance. The anomaly increases from 0.9 Gt/year in 1996 to 7.5 Gt/year in 2022 (Fig. 5a). The anomalies in the surface mass balance versus the 1979-2004 period remain at about -0.5 Gt/year between 1996 and 2022 (Fig. 5a). The anomalies in the total mass balance, $M$, therefore, increased by a factor of 6.6, from −1.3 Gt/year to -8.6 Gt/year, which is considerable compared to other glaciers in Antarctica[1]. In 25 years, the ice discharge of Berry Glacier has increased by 118 ± 8 Gt while the surface mass balance has decreased by 13 ± 15 Gt, that is, a cumulative mass loss of 131 ± 23 Gt (Fig. 5b), with 90% due to ice discharge (Fig. 5b). Despite its small size, Berry Glacier has therefore had an enormous influence on the mass loss of this region. Its mass loss is one third of the mass loss of the entire Getz Ice Shelf system when comparing our results with the estimates in[1].

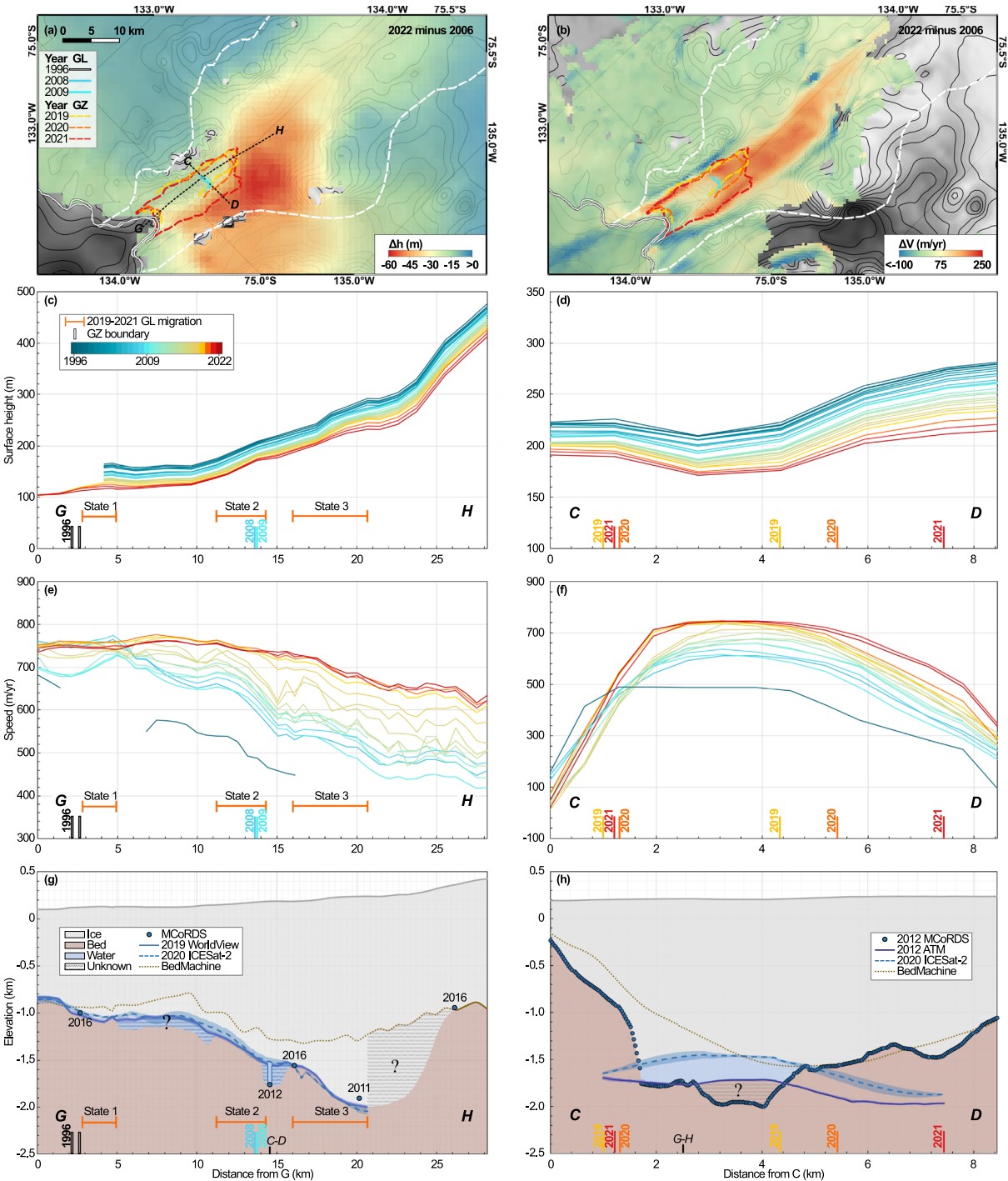

**Fig. 4 | Changes in surface elevation, *h*, and speed, *V*, of Berry Glacier, West Antarctica.** Changes in **a** *h* and **b** *V* from 2006 to 2022 with GLs in 1996 (white)/2008 (blue)/2009 (cyan) and IGZ boundaries in 2019 (yellow)/2020 (orange)/2021 (red); **c, d** *h* and **e, f** *V* along profiles G-H and C-D for 1996-2022; **g** *h* (grey solid) from WorldView (WV) DEM on Sep 6, 2019, ice draft (blue) deduced from WV (blue solid) and from ICESat-2 DEM for Jan 1, 2020 (dash blue) versus bed elevation from BMv3.7 (dash brown) along G-H. Blue circles are Multichannel Coherent Radar Depth Sounder (MCoRDS) data from 2011-2016; **h** *h* (grey solid) from Airborne Topographic Mapper (ATM) from Oct 27, 2012, ice draft (blue) from the ATM for Oct 27, 2012 (dark blue solid) and from the year 2020 ICESat-2 DEM (dash blue) versus bed elevation (dash brown) from BMv3.7 along C-D. Blue circles are MCoRDS data from Oct 27, 2012. Blue shaded areas in (**g, b**) are the uncertainty of the derived ice drafts. Blue stripey areas in (**g**) indicate regions where vertical motion from seawater intrusion is detected, but the bed depth remains unknown. White stripey areas in (**g**) denote regions outside the IGZ where flotation is not applicable and the BMv3.7 bed is not reliable. Brown stripey areas in (**h**) mark regions where the bed inferred from MCoRDS is not compatible with flotation from 2012 ATM and has been picked erroneously too deep.

## Implications for bed topography

We reconstruct the bed topography based on the minimum depth of ice that ensures flotation of ice over the zone of flexing during the tidal cycle, i.e., ice must be within a meter of flotation to allow ice flexing. The elevations of the ice draft deduced from the ICESat-2 and WV DEMs are consistent with each other (Fig. 4g). However, the calculated basal elevations are up to 800 m lower than those reconstructed in BMv3.7 along the profiles (Fig. 4g,h), with discrepancies reaching 1,300 m on the eastern side of the 2021 IGZ, where no radar profiles are available. The elevation of the BMv3.7 bed matches the thickness

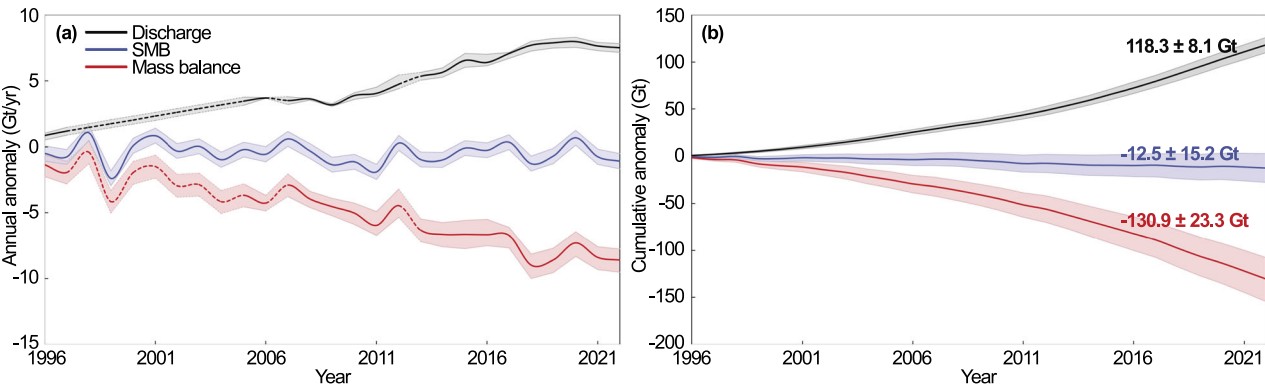

**Fig. 5 | Cumulative anomalies in ice discharge (*D*), surface mass balance (*SMB*), and mass balance (*M*) of Berry Glacier, West Antarctica from 1996-2022.**
**a** Annual anomalies in *D*, *SMB* and *M* with error bars in gigatons (10⁹ tons). Linear

interpolation is used to fill discharge gaps (dashed) in 1997-2005, 2007 and 2013; **b** cumulative anomalies in *D*, *SMB* and *M* for 1996-2022. Balance discharge is the average SMB for years 1979-2004, or 4.3 ± 0.2 Gt/year.

profiles of the MCoRDS only in the upstream part of the 2021 IGZ (Fig. 4g). We conclude that the thickness of the MCoRDS may have been misinterpreted in 2011. In fact, a close inspection of the radar echograms indicates weak bed echoes at the glacier center, i.e., a considerable amount of uncertainty in bed detection, i.e., the 2011 OIB Level 2 data are incorrect (Supplementary Fig. 3d). We infer a much deeper elevation of the bed, which is also consistent with the gravity gradients in that area[21]. Overall, we infer a much deeper bed than expected around state 3, narrow water columns at states 1, 2, and 3, a water column of several hundred meters between state 2 and state 3, and an unknown water column between state 1 and state 2. However, the thinning rate between states 1 and 2 suggests that the water column in that region should be on the order of several hundred meters.

Comparing the spatial distribution of the GL positions with the new bed topography along the G-H profile (Fig. 4g), we find that the GL migrated over a mostly retrograde bed during 1996–2021. A series of bed depressions between the three states may have created pockets capable of trapping seawater during the tidal cycle. We attribute the water trapping to the complex cavity shape, which involves the resistance to the outflow of the hydrological network and along-slope gravity gradients as modeled in ref. 22. In several InSAR data where the GL is detected in state 1 or 2, we observe fringes of opposite sense motion upstream, e.g., 11.5 cm (3.7 fringes of S1 DInSAR) in Fig. 3b, which suggests that seawater trapped in a prior tidal cycle is now able to leave the cavity. Similar fluctuations out of phase with the tide have been reported elsewhere and have been attributed to water trapping[15,16,23].

**Basal melting regime**

We calculate the ice shelf basal melt rate, $\dot{b}$, on the west Getz Ice Shelf including the IGZ for 2019–2022, from mass conservation using the ice velocity of MEaSUREs, the ice thickness deduced from hydrostatic equilibrium, the thinning rate of ICESat-2, and the surface mass balance data of RACMO2.3p2[24]. We use an Eulerian framework to apply mass conservation. To alleviate the deviations in the ice thickness from flotation, we apply a 4.5 km smoothing filter in the IGZ. We find the highest melt values at the entrance of the Berry IGZ and within the IGZ (Fig. 6). Within the Berry IGZ, which is 114 km² in area, the basal melt rates range from -30 to 90 m/year and average 67 ± 26 m/year. High values are predominantly found along steep retrograde slopes. Low values are found in flatter parts of the bed and near bed ridges.

During the past 25 years, a subglacial cavity has developed beneath Berry Glacier, with direct observations along the 2012 OIB track near state 2 suggesting a local cavity depth of up to 300 m (Fig. 4h, Supplementary Fig. 4b–d). The depth of the cavity away from the OIB transects is uncertain. The DInSAR signal supports the

presence of water intrusions, but does not constrain the cavity thickness. A time series of Landsat images confirms that bumps in surface height present in the earlier part of the record, which are indicative of ice grounded on a bumpy bedrock, disappeared during the retreat, to lead to smooth variations in ice elevation, which are consistent with flotation (Supplementary Video 1).

Conductivity temperature depth (CTD) data at the western Getz front show the presence of warm CDW near Berry Glacier as early as 1994, with an ocean temperature at 1,000 m depth around +0.2 °C (−0.1°C in 2000), increasing to +0.4 °C by 2007 (Supplementary Fig. 5b)[14,25], i.e., the ocean heat flux must have increased during that time period. The bathymetry on the continental shelf includes deep and wide channels that should facilitate the access of CDW to the IGZ of Berry (Supplementary Fig. 6). We delineate a thalweg along western Getz, from the ice front to the east of 134° W, which may provide access of CDW to Berry IGZ (Supplementary Fig. 5a)[21,26]. More recent CTD data indicate a continued warming trend of CDW along the western Getz Ice Shelf since 2007. Temperatures increased from +0.8 °C to +1.2 °C at 750 m depth in front of Dean Island between 2007 and 2018 (Supplementary Fig. 6). Although these stations are not located directly in front of Berry, they provide strong circumstantial evidence for widespread ocean warming throughout the western Getz. The warming trend has been attributed to improved Ekman pumping over the Siple Trough shelf break, which contributed to the presence of warmer CDW in front of the western Getz Ice Shelf in 2018 compared to 2000 and 2007, and also facilitated CDW inflow onto the continental shelf to increase the heat flux available at glacier grounding zones[25]. Together, these observations support the interpretation that CDW intrusions, modulated by shelf-break dynamics and channeled through topographic depressions, have played a central role in the retreat of Berry Glacier.

We compare the melt rates within 10 km downstream of the IGZ on the west Getz Ice Shelf for 2019−2022 (this study) with the inferred rates for 2003−2008[27] and 2010−2018[28] since we have data in this area for all three periods, but not inside the IGZ. Focusing on the area of coincident measurements, we find that the melt rate within 10 km of the Berry IGZ increased from 9.1 m/year in 2003-2008 to 14.9 m/year in 2019-2022. These values are higher than those for other glaciers, e.g., the DeVicq Glacier. The ocean temperature in front of DeVicq Glacier at 700 m depth increased from −1.2 °C in 2000 to +0.3° C in 2007, more than Berry, but unlike Berry Glacier, we find no obvious deep channel that would allow 700 m deep CDW to reach the IGZ of DeVicq (Supplementary Fig. 5). The difference in melt between the two glaciers increased from a factor of 1.3 × in 2003-2008 to a factor of 1.7 × in 2019−2022. The melt rates downstream of the IGZ of Berry are also the largest on West Getz Ice Shelf (Supplementary Fig. 7). The melt rates for Berry in 2019−2022 are 110% higher than the steady-state values

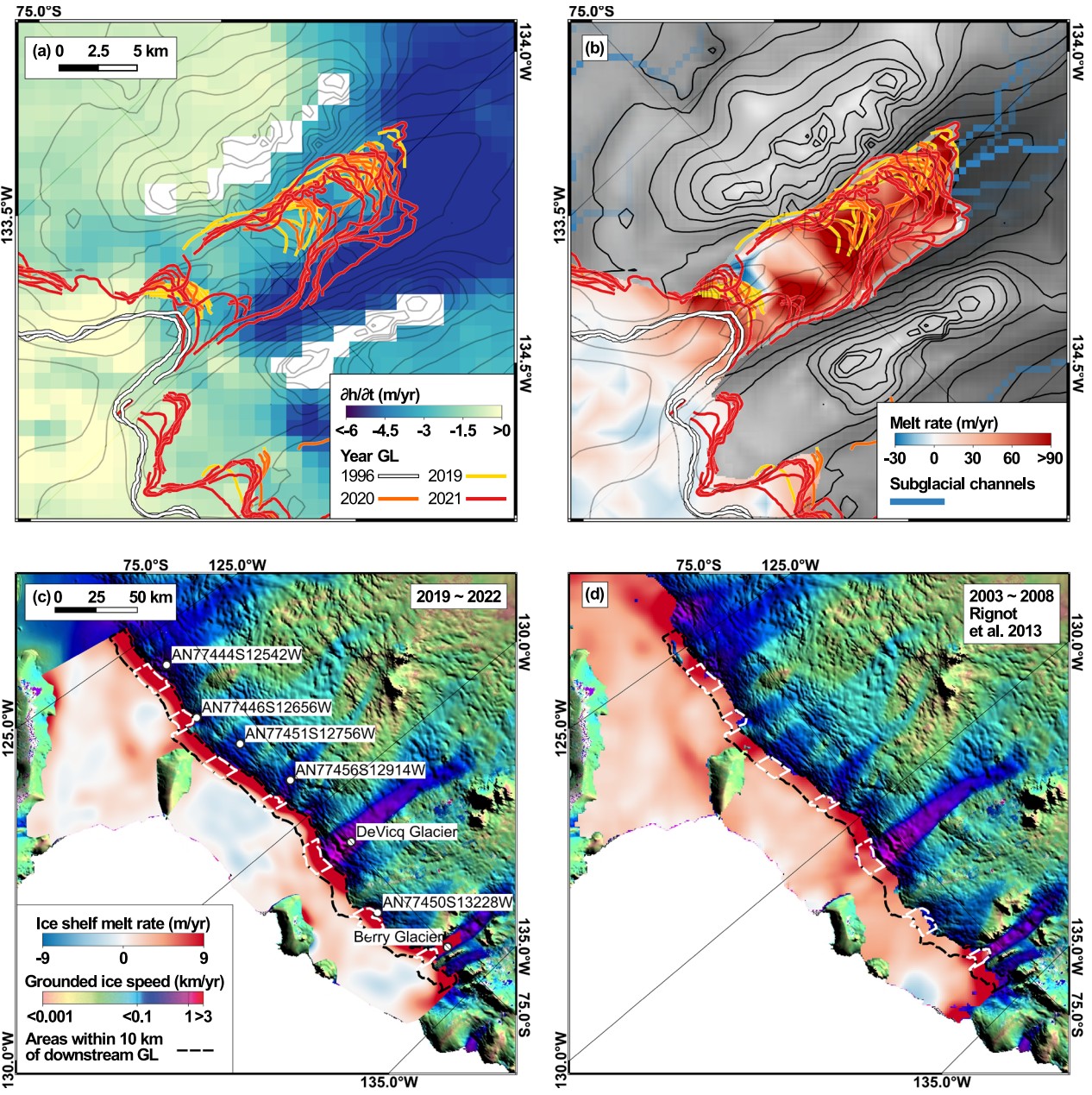

**Fig. 6 | Rate of change in surface elevation, $\partial h/\partial t$, and basal melt rate, $b$, of Berry Glacier, West Antarctica from Jan 1, 2019 to Dec 31, 2022.** **a** Surface thinning color-coded from -6 to 0 m/year with GL positions in 1996 (white) and 2019–2021 (yellow, orange, red); **b** basal melt rate overlaid on bed topography with 100-m contour levels and with subglacial channels from GlaDS hydrology model; **(c)** basal melt rates on the west Getz Ice Shelf for 2019-2022 from this study and **d** for 2003-2008 from[27]. Black dash line in (**c, d**) delineates area within 10 km downstream of IGZ. White dash line in (**c, d**) marks area of coincident measurements of the average melt rate within the 10-km wide region for each glacier.

calculated for the year 2007–2008, that is, assuming no thinning (Supplementary Fig. 7d), indicating that the ice shelf in this sector is 110% out of balance.

From the modeled tidal height of ± 1.2 m (Fig. 3d), we know that the average water in the IGZ cavity must be at least 0.6 m. If we assume that the cavity geometry tapers linearly from a maximum of 1.2 m at the entrance of the IGZ to zero at the inland limit, then given an IGZ area of 114 km², a water flux of $0.7 \times 10^8$ m³ in 12 hours would translate into a mass flux of $1.6 \times 10^6$ kg³/s. Assuming an ocean temperature of +1.2°C, inferred from 2018 CTD observations near Dean Island, the ocean thermal forcing, referring to the salinity-dependent and depth-dependent freezing point of seawater, is +3.6°C at 1000 m depth. With a heat capacity for seawater of 3850 J/kg/K, this thermal forcing translates into a heat flux of $2.8 \times 10^{10}$ W. If water intrusions flush the

entire cavity during each tidal cycle and 30 ± 10% of the available heat is used to melt the ice with a latent heat of fusion of $3 \times 10^5$ J/kg, we calculate 0.7 ± 0.4 Gt/year of melt or an average melt of 6 ± 4 m/year, which is ten times lower than the remote sensing estimate for 2019–2022. The large difference between the back-of-the-envelope calculation and the remote sensing results is not due to an underestimation of ocean thermal forcing or the efficiency of ocean heat at melting ice; it reveals or confirms that the thickness of the water column coming in and out of the Berry IGZ cavity is much greater than 0.6 m, possibly by a factor of ten.

While the relatively small ice volume of Berry Glacier limits its potential sea-level-rise contribution, our results carry important implications for ice sheet modeling in Antarctica and also reveal an enormous influence of this glacier on the mass loss of this part of West

Antarctica. Most ice sheet models do not account for short-term variability in the GL position and assume zero melt at the GL[2]. Our observations show that these assumptions do not apply to Berry Glacier. The 18 km-long IGZ experiences a high melt rate, higher than anywhere else on the Getz Ice Shelf. The melt rate has increased in recent decades, as a result of the enhanced Ekman pumping of CDW. The increase in melt forced the GL to retreat, reducing the basal resistance to flow, allowing the glacier to accelerate, which is conducive to thinning and further retreat. However, the complex shape of the newly formed cavity, which includes ridges, over-deepening, and areas of partial grounding (Supplementary Fig. 8), must have limited the amount of ocean heat that can flood the cavity, and in turn moderate the rate of glacier retreat. A coupled ocean-ice sheet model, as in[4], with a precise description of the cavity shape, would be needed to reproduce the observed glacier evolution.

More importantly, previous modeling studies[29–31] have shown that including seawater intrusions on a km scale in an ice sheet model increases the projections of glacier mass loss by up to a factor of two. The same conclusion would apply here in the case of Berry Glacier, that is, projections of its evolution will be higher when including seawater intrusions revealed in this study. In response to a warming of the ocean waters reaching the glacier, Berry Glacier has experienced an 18 km GL retreat, a 64% speed increase, and a mass loss of 131 ± 23 Gt, which is considerable for a single glacier. We recommend a more detailed characterization of the Berry cavity shape and continued measurements of ocean temperature and salinity to understand the physical processes that control heat delivery in the IGZ, melt rates, and retreat in more detail. Similar characterization should take place on other ice shelves in Antarctica.

# Methods

## Double-difference interferogram

We used ERS-1/2 (Supplementary Fig. 1a, b), ALOS-1/2 PALSAR (Supplementary Fig. 1c–h), S1a/b, CSK and RCM (Supplementary Fig. 1i–l) InSAR data to delineate the GL of Berry Glacier from 1996 to 2023. ERS-1/2 operated on a 1 day cycle in 1995-1996 during the tandem mission and on a 35-day cycle otherwise, at the C-band frequency (wavelength 5.6 cm), with a spatial resolution of 13 × 4 m, respectively, in the sloping range (cross-track) and azimuth (long-track). ALOS-1 PALSAR operates on the L band (wavelength 24 cm) with a 41.5° look angle off the nadir, 7.5 m × 4.1 m resolution (slant range × azimuth), a width of 70 km, and a repeat cycle of 46 days. ALOS-2 PALSAR operates at L-band (wavelength of 24 cm), has a 14-day repeat cycle, and in Stripmap Fine mode provides 10 m resolution over 70 km swaths. The S1 data are acquired at the C-band frequency (wavelength 5.6 cm), in a 6-day repeat cycle, in Interferometric Wide Swath (IW) mode, with a spatial resolution of 2.4 m in the slant range versus 13.9 m in azimuth[32,33]. CSK is a constellation of 4 × low-Earth orbit satellites operating in the X-band (wavelength of 3.1 cm) at a slant-range spacing of 3 m with a 16-day repeat cycle. The SAR images are formed by concatenating 3 × CSK STRIPMAP-HIMAGE overlapping frames, each 40 km × 40 km, at 3 m post in both azimuth and range directions[34]. RCM is a three satellite constellation that operates at C-band (wavelength of 5.5 cm) and provides a 4-day repeat cycle (12 days per satellite). The mission offers a wide variety of imaging modes, we focus on the high resolution 5 m (Stripmap) mode data at supported incidence angles between roughly 20–45°. We differ 3 (or 4) radar images to form a differential interferogram that reveals the differential tidal motion of floating ice during 3 (to 4) epochs. This approach enables us to separate the short-term vertical motion of the ice caused by tidal flexing from the constant horizontal motion of the ice[35]. The GL is located at the upstream end of a dense set of interferometric fringes (Fig. 2a, c, e). We manually map the GL in 93 × high-quality double-difference interferograms. When the fringe boundary is unclear or the fringes merge or split, we estimate the positive and negative errors based on the fringe pattern to show the farthest and

closest distances that the GL may reach. We define GL positions for years 1996, 2008, 2009 and 2023, and IGZs for 1996 and 2019–2021. The limited data availability for 1996 may lead to an underestimation of the IGZ length in 1996, but we detect no residual signal upstream of the 1996 IGZ, and the annual series of $\partial h/\partial t$ along the flowline indicates that it is unlikely for the GL to have reached state 2 before 2006. One fringe is a differential vertical displacement, $\Delta d = \frac{\lambda}{2\cos\theta}$, where $\lambda$ is the wavelength of the SAR and $\theta$ is the angle between the incident line of the radar and the vertical direction. Multiplying $\Delta d$ by the number of fringes, we calculate $\Delta SSH$ from the InSAR data.

## Tidal height calculation and correction

Floating ice flexes in response to variations in oceanic tides and atmospheric pressure. To quantify the tidal state, we use the CATS2008 model, which has a precision of 5 cm[19] (Fig. 2b, d, f). We calculate the correction of the inverse barometer effect (IBE) at a rate of 1 cm/hPa[36], using anomalies in mean sea level pressure from the fifth generation of atmospheric reanalysis of global climate (ERA5) by the ECMWF, with hourly outputs, at a horizontal resolution of 31 km[20]. We use the mean pressure for the period 2019–2021 as a reference pressure.

## Grounding zone length from hydrostatic equilibrium

If the ice is in hydrostatic equilibrium, the length of the IGZ is $\Delta L_{HE} = \Delta h[\beta + \frac{\rho_i}{\rho_w}(\alpha - \beta)]^{-1}$, where $\Delta h$ is the variation in the tidal height, the slopes of the surface $\alpha$ and the bed $\beta$ are determined along the thickness gradients averaged over several ice thicknesses, and $\rho_i = 917$ kg/m³ and $\rho_w = 1028$ kg/m³ are the density of ice and seawater, respectively[18]. To calculate $\alpha$ and $\beta$, we use a WorldView DEM from September 6, 2019. The slopes are calculated along the direction of the migration of GL on a length scale of 3 × ice thickness.

## Ice velocity and surface height data

The ice velocity is from MEaSUREs at a grid spacing of 450 m from 1996 to 2022[37,38]. We used changes in surface elevation from ATL14 (land-ice DEM) timestamped to January 1, 2020 and ATL15 (land-ice height change) between 2019 and 2022, and MEaSUREs ITS_LIVE Antarctic Grounded Ice Sheet Elevation Change between 1996 and 2018. ICESat-2 produces a foundational DEM at a 100-m post and quarterly elevation differences at 1 km resolution that document surface elevation changes with reference to the foundational DEM[39,40]. MEaSUREs products offer monthly ice sheet elevation changes derived from 5 × radar altimetry missions (Geosat, ERS-1 and -2, Envisat and CryoSat-2) and 2 × laser altimetry missions (ICESat and ICESat-2) at a resolution of 2 km[41].

## Ice mass balance calculation

Using $\partial h/\partial t$ along flowlines inferred from the elevation change products of ATL15 and MEaSUREs, we find that the IGZ reaches state 2 by 2006 and state 3 by 2012. We selected tracks from the OIB missions in 2016, 2012, and 2011 to serve as flux gates for 1996, 2006-2011, and 2012-2022, respectively, because the locations of the three OIB tracks overlap with the three states[42]. The OIB missions combine ATM altimetry and MCoRDS ice thickness. The ATM operates by emitting laser pulses toward the surface and measuring the two-way travel time, which is then converted into surface elevation data with an accuracy of a few centimeters. MCoRDS runs at low-frequency radar wavelengths (typically 150-600 MHz) to penetrate thick ice and detect internal layers and the ice-bed interface[43–47]. We calculated the ice thickness with the HE assumption at the designated flux gates for the period 1996-2022 by combining MCoRDS ice thickness with $\partial h/\partial t$ from MEaSUREs (1996-2018) and ICESat-2 (2019–2022). We apply the HE assumption only to regions where seawater intrusions are inferred at a specific time period. For the period 1996–2005, changes in ice thickness were equated directly with surface elevation changes, as the GL reached to state 2 around 2006. Between 2006 and 2011, the HE assumption was applied within the region from state 1 to state 2. From

2012 to 2022, HE was applied across the entire 2021 IGZ extent. By integrating these time-dependent thickness measurements with annual velocities in the direction perpendicular to the flux gates, we calculate annual ice discharge. We estimate the uncertainty in ice discharge by calculating its standard deviation from a central value in a region ± 1 km from the original flux gate.

The annual SMB for the Berry Glacier basin is from the Regional Atmospheric Climate Model Version p2.3 (RACMOp2.3)[48] with a resolution of 2 km with a 0.1 m/year error. The average SMB of the years 1979–2004, $SMB_{1979–2004}$, is used as a reference surface mass balance that would keep the glacier in a state of mass equilibrium, to calculate anomalies in ice discharge, $\delta D$, and SMB, $\delta SMB$. The total mass balance, $\delta M$ is $\delta SMB - \delta D$, which is independent of the reference period.

## Ice bottom and bed topography

We used the ICESat-2 DEM and the WorldView DEM to deduce the basal elevation of ice in the IGZ from flotation. The basal elevation of the ice $h_{bottom} = h - \frac{\rho_w}{\rho_w - \rho_i} h$, where $h$ is the elevation of the ice surface, and $\rho_i = 917$ kg/m³ and $\rho_w = 1028$ kg/m³ are the densities of ice and seawater, respectively. The WorldView DEM is generated from stereo imagery captured by Maxar's WorldView satellite constellation with spatial resolutions of up to 0.5 m and vertical accuracies of 3 to 5 m[49]. Along profiles overlapping with OIB tracks, we utilize ice surface elevation from ATM minus MCoRDS thickness to calculate the elevation of the ice bottom. We also calculated the elevation of the bottom of the ice from the ATM. The elevation of the DEMs is corrected with the EIGEN-6C4 geoid of BMv3.7. Firn air content (FAC) is removed from all DEM using the values in BMv3.7, which uses a firn densification model[50]. Although we apply FAC from BMv3.7 to account for the density difference between snow and ice, we acknowledge that the FAC in IGZ is uncertain, as firn models may not capture the complex densification processes that occur in these transitional environments[50].

Along profiles A-B, C-D, and C-F, the bed layers are questionable because the bed echoes are weak and incompatible with our calculations of flotation, which is equivalent to setting up a minimum ice draft (Supplementary Fig. 4a–c). Level 2 MCoRDS data for the year 2011 are not robust because the values read from the level 2 data do not match with the bed line picked in the radar map, which is why we find many 100 meters difference between our calculated bed and BMv3.7. The blue stripey areas in Fig. 4g mark the regions where vertical motion from seawater intrusion is detected, but the bed depth is unknown. The white stripey areas show regions outside the IGZ where flotation cannot be applied and the BMv3.7 bed is not reliable. In Fig. 4h, the brown stripey areas are regions where the bed inferred from MCoRDS is not compatible with flotation and, therefore, incorrect. The absence of clear bed echoes at that location confirms that the elevation of the bed was underestimated from the radar echograms.

## Ice basal melt rates

We apply an Eulerian approach to calculate the basal melt rates from mass conservation[24] using ATL15 products[39,40]. The melt rate is $\dot{b} = \dot{a} - \frac{\partial H}{\partial t} - (H(\nabla \cdot \mathbf{u}) + \mathbf{u} \cdot (\nabla H))$, where $\dot{a}$ is the surface mass balance, $H$ is the thickness of the ice, $\mathbf{u}$ is the velocity, $\nabla \cdot \mathbf{u}$ is the velocity divergence and $\nabla H$ is the thickness gradient. Since ICESat-2 only provides elevation change relative to a single DEM, rather than absolute surface elevations at different time steps, as required for a full Lagrangian analysis, the Eulerian approach is the only option to calculate basal melt rates. This approach requires spatial filtering of the results to mitigate the effect of heterogeneities in ice thickness and bending stresses. We used a spatial filter of 4.5 km, which approximately corresponds to the size of the bending zone discussed by[51]. Since the ice in the IGZ is not floating during the entire cycle, the melt rates calculated along the IGZ may be overestimated. Using DEMs separated by one year, we exclude points within a "limit of viability" zone where the ice was grounded at

the beginning of the integration period, equivalent to half the annual displacement (Fig. 6b). To evaluate the uncertainty in melt rate, we assume that the errors in the components of the mass balance equation are independent, unbiased, and follow a Gaussian probability distribution. We calculate an uncertainty of 21%, similar to Rignot et al. (2013) using an error in the velocity of 1% or 5 m/year, 3% in thickness or 30 m, 7% in $SMB$, and 20% in thinning rate.

## Data availability

The data generated in this study have been deposited in the Dryad database at https://doi.org/10.5061/dryad.k0p2ngfgt. The S1 data are provided by the European Space Agency at https://sentinels.copernicus.eu/sentinel-data-access. The CTD data supporting the findings of this study are available from the Met Office Hadley Center at https://climatedataguide.ucar.edu/climate-data/hadiod-met-office-hadley-centre-integrated-ocean-databaseand from the Korea Polar Data Center at https://doi.org/10.22663/KOPRI-KPDC-00000907.

## Code availability

The Python code used to calculate the basal melt rate is available through Zenodo at https://doi.org/10.5281/zenodo.12658904.

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

## Acknowledgements

This work was performed in the Department of Earth System Science, University of California, Irvine, and at Caltech's Jet Propulsion Laboratory under a contract with the National Aeronautics and Space Administration. H.C. and J.K. were funded by a NASA Earth and Space Science FINESST Fellowship (Grant 80NSSC20K1618 and 80NSSC21K1620). E.R., B.S., M.M., R.G., and P.M. were funded through the NASA MEaSUREs program (80NSSC23M0146). E.R. also received funding through the NASA Cryosphere Science program (80NSSC23K0177). E.C. was funded by a fellowship from the NASA Postdoctoral Program at JPL. This work contains ERS-1/2 data (1996) as well as modified Copernicus Sentinel data (2019), processed by the European Space Agency (ESA) and retrieved from the Alaska Satellite Facility (ASF). The authors thank the Italian Space Agency (ASI) for providing CSK data (original COSMO-SkyMed product ASI, Agenzia Spaziale Italiana (2020-2021)). RADARSAT Constellation Mission Imagery (c) Government of Canada (2023)-All Rights Reserved. RADARSAT is an official mark of the Canadian Space Agency. ALOS-1 PALSAR data (2008-2009) were provided by the Japanese Aerospace Exploration Agency (JAXA) and retrieved from ASF. The ALOS-2 data (2021) have been provided by JAXA EORC and were processed within the framework of the JAXA Kyoto & Carbon (K&C) Initiative.

## Author contributions

H.C. and E.R. designed the study, interpreted the data, and wrote the manuscript. H.C. carried out all data processing and created the figures. E.R., B.S., P.M., and E.C. generated the double-difference interferograms. M.M. upgraded and provided the customized BedMachine v3.7. R.G. helped with the basal melting calculation. J.K. helped with the OIB level 1 data interpretation. L.D. provided support with the CSK data. H.C., E.R., B.S., P.M., M.M., R.G., E.C., J.K., and L.D. contributed to the manuscript preparation.

## Competing interests

The authors declare no competing interests.
