## [Transparent Peer Review file · Nature Communications]

Rapid retreat of Berry Glacier, West Antarctica linked to seawater intrusions revealed by radar interferometry

Corresponding Author: Dr Hanning Chen

Version 0:

Reviewer comments:

Reviewer #1

(Remarks to the Author)

Summary:

This study uses a wide range of satellite observations from 1996-2022 to document changes in grounding line (GL) position, tidal grounding zone (GZ) dynamics, ice velocity, discharge and mass balance of Berry Glacier, which feeds 10% of the Getz Ice Shelf in West Antarctica. The authors report substantial increase in the length of Berry's ice grounding zone (IGZ), from 0.6 km in 1995/96 to 18 ± 0.9 km in 2020-21. Over the same period, they report changes in ice dynamics, including an estimated 17 ± 1 m/yr of thinning, a $64 \pm 5\%$ increase in ice flow speed, and a total mass loss of 131 ± 23 Gt. The authors present novel observations of short-term GZ dynamics within the Berry IGZ show that the GL migrates between three states, independently of tidal forcing, suggesting complex interactions between tides and GL movement that are not adequately captured in current ice sheet models. These short-term dynamics are proposed to facilitate seawater intrusions, driving the increased basal melt, thinning, and ice discharge. Furthermore, the authors propose that the bed topography within the Berry IGZ is up to 1,300 m deeper than indicated by MCoRDS airborne radar.

General Comments:

This work has impressive scope, integrating a wide range of satellite datasets over Berry Glacier – a region that has not previously been the subject of such a comprehensive study. I believe the results of the study will be of interest to the community, particularly regarding the novel observations of tidal seawater intrusions and their potential impact on basal melt rates within grounding zones. However, I have several significant concerns that should be addressed by the authors before publication. These concern the lack of robust uncertainty estimation or discussion of limitations, as well as certain aspects of the methodology and data interpretation that may affect the validity of the conclusions. It is crucial that the distinction between observations and interpretations is made clear throughout the manuscript. Additionally, the text and figures require refinement in many places to bring them up to publication standard. Below, I outline my major concerns below, followed by a list of specific suggestions (in attached file) for improving text clarity, figures, code and datasets.

Originality and Significance:

The study provides new insights into the dynamics of the IGZ at Berry Glacier and the first published observations of changes in ice velocity, discharge and melt rates over the past 25 years. However, it should be noted that Berry Glacier is relatively small in the context of the Antarctic Ice Sheet. Thus, while the findings are significant for this setting and contribute to improved process-level understanding, their broader implications for ice sheet modeling and sea level rise projections are potentially more limited. I recommend enhancing the discussion of these broader implications, addressing—for example—how findings at Berry compare to those from other Antarctic IGZs and how they might contribute to modeling improvements on an ice-sheet-wide scale.

Treatment of uncertainties:

There is an insufficient and inconsistent treatment of errors and uncertainties throughout the analysis, including in the main text, methods and attached dataset. I strongly recommend that the authors include a more comprehensive discussion of the uncertainties and errors in each part of the analysis and how these propagate through the calculations. I recognize that quantifying some of these uncertainties may be challenging; nevertheless, an open discussion of these limitations is essential to avoid potentially misleading readers or overinterpreting the results.

Robustness of Conclusions:

I have some concerns about the validity of some of the data interpretation that form the basis of the paper's main conclusions. I would like to see a response from the authors providing further clarification or justification of these.

1. Long-term GL retreat 1995-2021

The authors report that the length of the Berry IGZ increased from 0.6 km in 1995/96 to 18 ± 0.9 km in 2020-21. The 1995/96 IGZ length seems to be based on data from just two ERS1/2 interferograms, compared to >15 interferograms for 2020-21. I appreciate that there is limited data availability from 1995/96, but I wonder whether the reported shorter IGZ length in the 1990s may be influenced by the tidal sampling resolution. If the two ERS1/2 interferograms only captured the GL when it was in its Stage 1 position, it's possible that any inland migration of the GL during other tidal phases was missed. This could lead to an underestimation of the IGZ length during this period, and therefore an overestimation of the long-term change by 2020-21. I recommend that the authors discuss this potential bias in the interpretation and clarify how tidal sampling might impact the results (1-2 sentences would suffice). Providing the modeled differential tide and maximum sampled tide for each of the ERS1/2 SAR scenes in both Supplementary Figure 6 and in the data spreadsheet (as for the Sentinel-1 and COSMO-SkyMed data) would provide valuable context.

2. Calculations of ice thickness using HE

The authors compare the GL retreat at Berry to observed changes in velocity and ice elevation from 1996-2022 and estimate how this translates to changes in ice thickness, discharge and total mass balance. I am concerned about the validity of using hydrostatic equilibrium (HE) to convert ice elevation change observations to thickness changes. Additional justification of this approach is needed, along with a more open discussion of the associated limitations and uncertainties in the main text. While using HE for this calculating ice thickness is appropriate for fully floating ice, the InSAR data indicate that the ice in the Berry IGZ is grounded for a significant portion of the tidal cycle, with an average depth of seawater intrusions of ~0.6 m. This implies the ice within the IGZ cannot be in hydrostatic equilibrium, casting doubt on the validity of ice thickness values derived from this method, and, by extension, the estimates of thinning rates, ice discharge, basal melt rates and bathymetry. For example, the authors' interpretation that a 24 m surface elevation decrease in the IGZ translates to 222 m of thinning (L101-102) is likely to be a significant overestimation.

It is well-documented that satellite-derived estimates of ice thickness and basal melt rate within GZs are subject to large uncertainties due to the breakdown of the hydrostatic assumption (e.g., Chuter & Bamber, 2015; Griggs & Bamber, 2011; Kim et al., 2024), which often leads to these regions being masked out (e.g., Griggs & Bamber, 2011; Paolo et al., 2015; Wilson et al., 2017) or interpolated across (e.g., Fretwell et al., 2013; Morlighem et al., 2020). While I note that some attempt has been made by the authors to address this (e.g. L150-152; 296-298), the discussion remains too vague and does not adequately address the impact on the key findings.

3. Refinement of bed topography

The authors propose that the bed topography within the Berry IGZ is up to 1,300 m deeper than measurements from MCoRDS airborne radar and in BedMachine v3.7 (which interpolates between sparse radar measurements). However, I do not think that the evidence presented here is strong enough to support this conclusion; I suggest the authors either reconsider this interpretation or provide a more robust justification, clearly communicating any limitations. Specific concerns include:

- Methods description:** The description of methods used to reconstruct bed topography lacks sufficient detail, both in the main text (L125-127) and methods (L278-279). I recommend expanding the description to clarify the approach, including details on the geoid and specifying the areas or flowlines over which bed topography was calculated.
- Use of HE/flotation:** If the new bathymetry estimates in the IGZ are based on ice thickness calculated using HE/flotation, they are subject to the same uncertainties outlined above. Prior studies have demonstrated that ice thicknesses calculated in this way close to the GL tend to overestimate values compared to radar measurements due to the breakdown of HE (e.g. Chartrand & Howat, 2023).
- Radargram interpretation:** The authors propose that the bed depths from the MCoRDS radar in the Berry IGZ have been misinterpreted as too shallow (L132-134). However, upon reviewing Supp. Figure 2 (noting that I am not a radar specialist), I find this interpretation unconvincing. While weak bed echoes at the glacier center may well indicate higher uncertainty in bed depths, the picked bed depths (red dashed lines) in all four radargrams look reasonable and there is little indication of any deeper echoes at the depths suggested by the authors. Radar is generally considered most reliable way to measure ice thickness/bed depth in these regions, and it seems more plausible that the HE-based calculations in this study might overestimate thickness/depth compared to the radar. Labelling specific features in the radargrams that support the deeper bed interpretation would be helpful, as well as providing a comparison to data from the gravity inversion mentioned in L136.
- Clarification of the 1300m depth difference:** The origin of the 1,300 m value is unclear. In Figure 3(g-h), the difference between the BedMachine bed elevation and the author's new calculations appear to max out at around 1,000 m. Since this 1,300 m value is presented as a key result in the abstract, its basis should be clarified in the text.

4. Long-term melt rates / basal melt regime

The text in this section (L146-192) needs refining; it is currently difficult to follow the specific areas and time periods for which melt rates have been calculated. Clear distinctions are needed between discussions of: (a) Melt rates within the Berry IGZ (2019-2022); (b) Melt rates within a zone 10 km seaward of the IGZ, compared to neighboring glaciers and previous measurements; and (c) Predicted melt rates within the Berry IGZ due to tidal seawater intrusions

Again, my concerns about the use of HE to calculate thinning rates within the IGZ alongside the lack of uncertainty quantification make me skeptical about the validity of the derived melt rates. The authors briefly mention using 4.5 km spatial smoothing to mitigate deviations from hydrostatic equilibrium (L150-152), but further detail is needed to explain what was done and how this reduces uncertainty. As I understand, the authors have interpreted the formation of a >100m thick cavity

beneath the Berry IGZ over the past 25 years (L156), reaching depths up to 300 m near State 2. This implies the whole region is afloat, which starkly contradicts the InSAR GL results which show the whole IGZ is ephemerally grounded with depths of seawater intrusions averaging just 0.6 m. Clarification of this discrepancy is important.

5. Tidal seawater intrusions

The estimates of melt rates due to tidal seawater intrusions (L181-192) appear to rely heavily on assumptions and values that are either poorly constrained or lack clarity on their uncertainties. This undermines confidence in the results and may mislead readers, especially if the values are subsequently used for in models. It is recommended that the authors clearly outline all assumptions, cite where the values are taken from (note specific comments below about the CTD data), and acknowledge the scale of the limitations.

Diversity, Equity and Inclusion:

I would like to draw attention to the lack of female representation among the nine co-authors, which might be something the authors or the journal could consider addressing in future to foster more diversity in scientific collaborations.

(Remarks on code availability)

The Zenodo link points to code used to compute basal melt rates at Thwaites, originally from a different paper (Gadi, 2024). I am not sure of the specific journal guidelines on this, but I suggest this should either be appropriately cited alongside the dataset, or specific code used for this work at Berry should be provided.

No code is provided for the other calculations described in the methods, including ice bottom/bed topography, ice mass balance, GZ length or tides. This should be included to ensure full transparency.

Reviewer #2

(Remarks to the Author)

A note in advance of this review: I am focusing on the physical oceanography part of the manuscript "Rapid retreat of Berry Glacier, West Antarctica from satellite radar interferometry and other data" by Chen et al. have not commented on the remote sensing and glaciology aspects, since these are outside of my area of expertise as a physical oceanographer. I am referring quite heavily to my own work as a consequence of the scarcity of oceanographic observations in the area and have thus chosen to write this review non-anonymously.

Karen Assmann

Overall I think the manuscript contains exciting, new results, but the line of argument around the connection between the GL retreat of Berry and changes in CDW temperatures needs revision and refinement to make it convincing.

Generally, the manuscript needs some polishing. I suspect that shortening the text to make it fit within the Nature Communications format is the culprit here. This applies especially to the introduction that needs to be reworked – see my specific comments.

There are two main points in the manuscript. One is that you can identify tidal variability in the movement of the glacier and that this likely gives the warm CDW on the continental shelf access to the grounding line zone leading to high melt rates. The second one is that the grounding line of Berry has retreated rapidly over the past 25 years due to that CDW access.

My main problem here is that your line of argument focusses on the presence of CDW and not how it may have changed. There isn't much CTD data available, but what there is could be used better:

Show the temperature profiles that you are using and don't just leave the reader to eye-ball them from Jacobs et al. 2013.

Also give data sources for the CTD data, especially the mysterious 2018 CTD that you base your heat flux calculation on.

Use these temperatures and the position of the thermocline vs the associated GL positions and cavity shapes to discuss the implications for basal melt.

CDW access to the deep basin in front of Berry is most likely across the shelf break and not from the east on the shelf. The sill depth is shallower than for Siple Trough (see Jacobs et al. Oceanography 2012) and the off-shelf CDW core actually sits below the sill depth. This implies that it is likely even more episodic than we have shown for Siple Trough and even more dependent on winds along the shelf break and Ekman pumping (Assmann et al. 2019).

The implication of this is that you have a mechanism that explains why CDW access may have changed between 1994 and 2021: If you compare your mass balance time series in Fig. 4a with our Ekman pumping time series in Assmann et al. 2019 Fig 4f they agree rather well. You could calculate a similar time series for the shelf break in front of Berry, but it is unlikely that it would be much different because of the scales of atmospheric variability. Just like we argued that the enhanced Ekman pumping over the Siple Trough shelf break led to warmer CDW in front of the western Getz in 2016 & 2018

compared to 2000 and 2007 that argument could be used in this study as well.

Specific comments:

Line 42: "affected" not "effected".

Line 45: You are implying that changes in GL position affect sea level, because of the addition of seawater by floating ice. Floating ice shelves are hydrostatically balanced, i.e., they don't affect sea level. What does affect is the mass flux across the grounding line and the mass loss from the ice shelves by basal melting and calving that adds freshwater, i.e., volume to the ocean. Just remove this sentence.

Line 50: This sentence is very out of context, move to last paragraph in the introduction where you introduce the location.

Line 52-53: "due to ocean tides" not "changes in oceanic tides". Tides imply variability.

Line 56-67: For my taste this is too technical for an introduction. I got lost in the detail and wasn't sure why you are using all of these satellite data sources.

Line 66: You haven't given any geographical context, i.e., the reader has no idea where or what Berry is. I assume this is result of shortening the text to fit within the Nature comms format.

Line 71: Change “undiluted warm, salty CDW” to “warm CDW”. You don’t need salinity to make your point and I would remove it to simplify the story. CDW access on the western side of Getz is intermittent, so the CDW that makes it to the ice shelf front is anywhere from part of the thermocline to part of the CDW core (see Assmann et al. 2019).

Line 147-152: Duplicates the methods section, so you could remove this.

Line 161-162: Remove the reference to salinity. This is superfluous to your main storyline. Either write “mean bottom temperature of...” or change to something like “bottom temperature exceeding 1 C” to match the CTD sections in Jacobs et al. 2013 and sections and mooring records in Assmann et al. (2019).

Line 162-165: see comment about Supp. Fig. 4a and general comment about CDW access from the shelf break.

Line 165-167: This is the only mention you have in the entire article about changes in ocean temperatures and it is too hand-waving to support your main point. Show the actual temperature profiles in a figure, because it is hard to pin-point Berry in the Jacobs et al. (2013) figures.

Line 181-185: I am not sure why you use 0.6 m as a thickness when you draw in 1.2 m worth of water volume over a tidal cycle.

Line 186: The tide is diurnal at Berry (and generally in the Amundsen Sea), i.e., half a tidal cycle is 12 hours not 6 hours – see your Fig. 1.

Line 187: Where is your 2018 CTD from? Add a source for all CTD data you use to your data statement. This is especially important for this 2018 CTD since you use the temperature for your heat flux calculations.

Line 190: I like the Gadi et al. (2023) reference a lot and agree that not all heat in the inflowing seawater is used to melt ice, but I interpreted the 30% statement in their study that 30% of the ice flux across the grounding line was melted in the GZ and FZ.

Fig 1: Even with the improved resolution, panels in Fig 1 are still very small, this applies particularly to the top left insert in panel a.

Supp. Fig. 4 a: Add the ice shelf front to panel and extend the figure to the continental shelf break to show the channels giving the CDW access. In my experience, the bathymetry on the continental shelf around the western Getz isn’t very detailed or reliable, because I don’t think there is more multibeam data than shown in Nitsche et al. (2007). I would qualify the statement you make about these channels in lines 162-165.

Supp. Fig. 4 b & c: I do like the cavity sections, because they show nicely how the Berry GZ has moved to a retrograde slope, while the DeVicq GZ sits just at a small rise. I also suspect that bed shape is more responsible for the different reactions of the two glaciers rather than CDW temperature.

The temperature contours in panels b and c imply a thermocline at constant depth in front of Berry. I would have preferred a profile insert that shows that this is a point measurement. Showing would also give you the opportunity to show profiles for different years and to make your point about the CDW changes.

Supp. Fig. 5: I do wonder if it would be worth to move some of the panels into Fig. 5, because they illustrate the exceptional reaction of Berry within the region and give context to your discussion of DeVicq glacier which feels somewhat stand-alone in the manuscript.

(Remarks on code availability)

Reviewer #3

(Remarks to the Author)

Review for ‘Rapid retreat of Berry Glacier, West Antarctica, from satellite radar interferometry and other data’

In this study, Chen et al. used DInSAR interferograms generated from different satellite data to map both the short-term and long-term grounding line migrations in Berry Glacier located in the Getz Ice Shelf from 1996 to 2021. They discovered a short-term grounding line migration over 18 km and claimed that the glacier bed is up to 1300 m deeper than the BedMachine data. The methods used in this study are standard and not novel, particularly for research focused on grounding line mapping using DInSAR. The identification of a large ‘Ice Grounding Zone’ in Berry Glacier is also not new, as it was previously discovered by Mohajerani et al (2021). The paper does not offer significant advances in our understanding of either short-term grounding line migration or long-term changes related to ice dynamics, as these concepts have already been extensively explored and established in previous studies. Furthermore, the interpretation of bed topography and the calculation of short-term grounding line migrations require further consideration and improvements.

Major Comments:

The finding of the large tidally-influenced grounding line migration in Berry Glacier is not new, Mohajerani et al (2021) have already found grounding line changes in Berry Glacier to be about 15 km (<https://www.nature.com/articles/s41598-021-84309-3>). Given this, I feel this paper only provides a more detailed calculation building upon the work of Mohajerani et al (2021), and its main conclusion lacks novelty. Especially, Mohajerani et al (2021) have analyzed the entire Getz Ice Shelf. I am curious why the authors in this study chose to focus solely on a single glacier within the same region. Including all glaciers in Getz Ice Shelf would allow for a comparison of grounding line changes across glaciers that share similar oceanography settings, this would result in a more comprehensive study. Focusing on just one glacier significantly limits the scope of this research.

I agree that short-term grounding line migrations have not been adequately incorporated into ice sheet modeling. However, this phenomenon has been well-documented in Antarctica in recent years using a similar DInSAR approach and datasets to those used in this study, with some examples listed below. While I acknowledge the efforts presented in this paper, I feel that

its conclusions do not significantly advance our understanding of short-term grounding line migrations. Instead, the study only adds incremental evidence to the existing body of literature.

- Milillo 2017: <https://agupubs.onlinelibrary.wiley.com/doi/full/10.1002/2017GL074320>
- Milillo 2019: <https://www.science.org/doi/10.1126/sciadv.aau3433>
- Brancato 2020: <https://agupubs.onlinelibrary.wiley.com/doi/10.1029/2019GL086291>
- Mohajerani 2021: <https://www.nature.com/articles/s41598-021-84309-3>
- Milillo 2022: <https://www.nature.com/articles/s41561-021-00877-z>
- Chen 2024: <https://agupubs.onlinelibrary.wiley.com/doi/10.1029/2022GL102430>

The comparison between 1996 IGZ and 2020-2021 IGZ needs more care. In 1996 you only have two interferograms, it is very likely due to limited DInSAR data acquisition during that time, this may have resulted in incomplete capture of short-term grounding zone migrations. While in 2020-2021, there is a greater availability of data, enabling greater sampling and a more comprehensive imaging of the entire IGZ. Please address this in the analysis.

Figure 2a shows the spatial density of different grounding line positions obtained in this study, capturing three different states. However, a limitation of this approach is the lack of consideration for temporal sampling. If the GL acquisitions are concentrated around same tidal phases, the resulting spatial distribution map may be biased. In addition, the gap between State 2 and 3 is too short (~1 km), making this classification unreliable.

It is interesting that the authors found the bed topography underneath Berry Glacier to be 1300 m deeper than the BedMachine. This is useful information that can be used to update the BedMachine dataset. However, I am not particularly surprised that the BedMachine dataset is inaccurate in some of the glaciers due to a lack of observations. Additionally, it is unclear how this discovery of deeper bed topography changes our current understanding of the grounding line dynamics and regional mass imbalance.

The discussion about seawater intrusion is unclear. In Line 139: the authors claim that 'A series of bed depressions between the three states must trap seawater...' I feel this claim is not well supported and more evidence is needed to make sure this statement is scientifically correct. Although the authors claimed to detect the reverse fringes of opposite motion upstream to indicate the trapped seawater, there are not enough details on how the water extent is mapped in Figure 3 and how the presence of trapped seawater changes in sync with grounding line migrations.

Mismatching patterns in ice dynamics: The dhdt in the west flank is much lower than the main glacier trunk where there is an over-deepened retrograde bed slope, this does not match the basal melt rate patterns in Figure 5 and the ice velocity changes in Figure 3. Based on MISI theory, we would expect to see matching patterns of ice thinning, grounding line retreat, as well as ice speed up along the overdeepened bed. The authors, however, only provide one comment on this mismatching pattern in Line 114-115: 'The pattern of thinning does not coincide with the pattern of speed up as the retreat expands laterally along a tributary'. This explanation is too simplistic and misleading. Please provide a more detailed explanation.

Specific Comments

Line 51-55: the jump from basal resistance associated with the grounding line to the lack of consideration of short-term grounding line migration in glacier modeling feels out of place here. Please consider talking about the importance of short-term grounding line and its observations across Antarctica first.

Line 57-59: the accuracy of GL mapping in altimetry and DInSAR needs references

Line 97: 'GL is predominantly in state 1 in 1996...' also depends on the number of DInSAR interferograms and the tidal samplings.

Line 129-130: 'up to 1300 m lower than those reconstructed in BMv3.7', could you explain more how this number is obtained in the figure or in the main text?

Figure 1: please increase the size of the inset map of Getz ice shelf in subplot 1a, it is difficult to see the details in its current format

Figure 3: please avoid using rainbow colormap in line plots 3c/d/e/f, it is hard to distinguish between different years, perhaps a sequential colormap is better. In 3g and h, please clarify the question mark annotated on the figure in the caption, this part is also unclear from the main text

Line 151: why 4.5 km?

Line 160: could you map or annotate the boundaries of this surface elevation bumps in the satellite images in Supplementary Video 1? It is difficult to distinguish this feature in the current video.

(Remarks on code availability)

I have not reviewed the code due to the above reasons.

Version 1:

Reviewer comments:

Reviewer #1

(Remarks to the Author)

I commend the authors for their thorough revision of this manuscript, and I agree that it is now significantly improved! The title is much improved and the majority of my concerns have been addressed, regarding the text, methods descriptions, and the treatment of uncertainties. The figures are in much better shape too. I have one major remaining concern regarding the discussion about the cavity formation (detailed below), and a few additional minor comments / suggestions that the authors may like to consider before final submission. Once addressed, I would recommend this manuscript is acceptable for publication.

I have attached my comments in a separate PDF file.

(Remarks on code availability)

I have not re-reviewed the code but the authors have made no changes.

Reviewer #2

(Remarks to the Author)

In general, the whole manuscript reads more balanced and is greatly improved after the authors incorporated mine and my fellow reviewer's comments.

The discussion of the influence of the CDW and its recent changes is now much better. Especially Supp Figs. 4 and 6 reflect the hydrography and data base much better.

As a minor revision before final acceptance of the manuscript I would like to see some re-organisation of the discussion of the influence of CDW in the main manuscript.

General comment:

Line 178-189 and 208-214 which contain the main discussion on CDW are somewhat repetitive and would benefit from merging and re-ordering. I suggest doing this following your very nice figures Supp. Fig. 4 and Supp. Fig. 6.

Supp. Fig. 4 shows that there are direct observations of CDW at the ice shelf front of both Berry and duVicq with some warming between 1994 and 2007. Use these to discuss access of the CDW to the GL and the importance cavity shape.

Then use Supp. Fig. 6 to take in the more recent observations and the CDW warming to present day. Since these are not located directly at Berry, this is the place to discuss CDW pathways on the continental shelf and the enhanced Ekman pumping at the shelf break as the mechanism that controls CDW access to the continental shelf and thus the warming. It is worth mentioning that the CDW at Berry may not have reached 1.2 C in 2018, since CDW there appears to have been cooler than at Dean in the earlier years.

Specific comments:

l.181: remove "because they are deep enough" Also since this refers to the bathymetry on the continental shelf outside of the ice shelf cavities that figure reference should be Supp. Fig. 6.

l. 208-209: "The ocean temperature on the western Getz Ice Shelf" should be "The ocean temperature along the western Getz Ice Shelf"

l. 236-238: This sentence is unclear. Re-write.

Supp Fig 6: Remove the CTD stations east of Siple. CDW there doesn't have direct access to your region of interest and this would make the figure easier to read.

(Remarks on code availability)

Reviewer #3

(Remarks to the Author)

Thank you for addressing my comments and revising the paper. I only have some minor suggestions below.

Line 71: Please consider rephrasing the sentence into 'are in contact with warm waters coming from Circumpolar Deep Water that ...'

Line 72: The mention of ice shelf meltwater feels out of place here. I suggest adding a statement on the implications of ice shelf basal melting.

Figure 1a: It's difficult to discern the black arrow associated with Berry Glacier from the background ice velocity map. I suggest changing to a different color. The same issue applies to the star marker at the tide location. Also, the star marker is very small.

Line 78: 'but there is no residual signal upstream of state 1 in 1996', please add a reference to Figure 7.

Line 83-86: Please specify which bed topography product you are referring to.

Line 172: Please rephrase 'several 100 meters'

Line 194 - 201: I can't tell changes in basal melt rates in different time periods or different locations mentioned in this paragraph directly from Figure 5, as the current colorbar of basal melt rate has a limit of [-6,6] m/yr. Please revise the figure to improve the illustration of basal melt rate variations.

Line 204: Consider removing the repetitive 'that is' in this sentence

(Remarks on code availability)

Version 2:

Reviewer comments:

Reviewer #1

(Remarks to the Author)

Response to Revisions (Round 2)

Thank you to the authors for their thoughtful responses to my comments and for the efforts made to address these in the revised manuscript. I am satisfied with their responses and with all changes made. The additions to both the text and figures have greatly strengthened the manuscript. Congratulations to the authors.

I have two very minor residual comments that the authors may wish to consider before publication:

- L140 - Missing unit on " 13 ± 15 " for the SMB.
- Figure 3 – Thank you for clarifying these points in your response and the revised manuscript. To avoid any potential confusion, I suggest using a different color for the blue striped regions (locations where vertical motion from seawater intrusion is detected) and the blue shaded regions (ice-bottom uncertainty bounds). At present, the two blues are difficult to distinguish, particularly where they overlap.

(Remarks on code availability)

I did not re-review the code, as I do not believe any changes have been made since the first round of reviews.

Reviewer #2

(Remarks to the Author)

I am satisfied with the author's response to my comments and the resulting edits in the manuscript. I hereby recommend publication of the manuscript.

(Remarks on code availability)

Dear Editor, Reviewers,

We are very thankful for the reviewers' comments. In response to these, we have extensively revised the manuscript. However, the impact of the revision on the results is minimal. We believe that the new version is significantly improved. We hope that the reviewers will find the revised manuscript acceptance for publication.

On behalf of the co-authors – Hanning Chen.

Reviewer #1

1. **Originality and Significance:** The study provides new insights into the dynamics of the IGZ at Berry Glacier and the first published observations of changes in ice velocity, discharge and melt rates over the past 25 years. However, it should be noted that Berry Glacier is relatively small in the context of the Antarctic Ice Sheet. Thus, while the findings are significant for this setting and contribute to improved process-level understanding, their broader implications for ice sheet modeling and sea level rise projections are potentially more limited. I recommend enhancing the discussion of these broader implications, addressing—for example—how findings at Berry compare to those from other Antarctic IGZs and how they might contribute to modelling improvements on an ice-sheet-wide scale.

Response: Yes, Berry Glacier is a small glacier, with a low potential for sea level rise but its evolution illustrates the fundamental physical processes operating in the Antarctic environment that - in the presence case - led to the formation of a gigantic ice grounding zone and a rapid retreat along a deep trough. The magnitude of the retreat, the size of the cavity, the magnitude of the melt rates, and the bathymetry are relevant aspects to a broader type of study of glacial retreat in a warmer ocean. Berry has also one of the longest IGZ in Antarctica, as first noted in Mohajerani et al. (2022). There are others: Barnett Glacier, Victoria Land; Vanderfjord Glacier, Wilkes Land, Lambert/Mellor/Fisher, Jutulstraumen, etc. In the revision, we discussed the broader implications of this study at Lines 226-240.

2. **Treatment of uncertainties:** There is an insufficient and inconsistent treatment of errors and uncertainties throughout the analysis, including in the main text, methods and attached dataset. I strongly recommend that the authors include a more comprehensive discussion of the uncertainties and errors in each part of the analysis and how these propagate through the calculations. I recognize that quantifying some of these uncertainties may be challenging; nevertheless, an open discussion of these limitations is essential to avoid potentially misleading readers or overinterpreting the results.

Response: We documented the uncertainty of the grounding line retreat, the time series of ice velocity, surface height, and inferred bed topography, which led us to find that some MCoRDS data were not correctly interpreted. We double checked the discussion of the uncertainty of the melt rates.

3. **Robustness of Conclusions:** I have some concerns about the validity of some of the data interpretation that form the basis of the paper's main conclusions. I would like to see a response from the authors providing further clarification or justification of these.

1) Long-term GL retreat 1995-2021

The authors report that the length of the Berry IGZ increased from 0.6 km in 1995/96 to 18 ± 0.9 km in 2020-21. The 1995/96 IGZ length seems to be based on data from just two ERS1/2 interferograms, compared to >15 interferograms for 2020-21. I appreciate that there is limited data availability from 1995/96, but I wonder whether the reported shorter IGZ length in the 1990s may be influenced by the tidal sampling resolution. If the two ERS1/2 interferograms only captured the GL when it was in its Stage 1 position, it's possible that any inland migration of the GL during other tidal phases was missed. This could lead to an underestimation of the IGZ length during this period, and therefore an

overestimation of the long-term change by 2020-21. I recommend that the authors discuss this potential bias in the interpretation and clarify how tidal sampling might impact the results (1-2 sentences would suffice). Providing the modelled differential tide and maximum sampled tide for each of the ERS1/2 SAR scenes in both Supplementary Figure 6 and in the data spreadsheet (as for the Sentinel-1 and COSMO-SkyMed data) would provide valuable context.

Response: Yes, because we have only two interferograms, the width of the IGZ in 1996 is not as well determined as in later years. Yet, there is also several lines of evidence indicating that the GL was at state 1 in 1996. First, we see no residual signal upstream of the state 1 IGZ in the two 1996 interferograms. In the more recent interferograms, there is always a residual signal upstream, between state 1 and 3, that indicates that during some part of the tidal cycle, there is residual vertical motion in that region (Fig. 1). It is unlikely that seawater intruded farther than state 1 in 1996. Furthermore, the glacier thinned considerably since 1996 (11 ± 1 m/yr). Ice deviated from flotation by a few meters at state 2, which the grounding line reached in 2006. See Lines 277-280. The 1996 IGZ could not have extended to state 2. As requested, in the revision, we also added information about the modeled differential tide for ERS1/2 SAR in Supplementary Fig. 7.

2) Calculations of ice thickness using HE

The authors compare the GL retreat at Berry to observed changes in velocity and ice elevation from 1996-2022 and estimate how this translates to changes in ice thickness, discharge and total mass balance. I am concerned about the validity of using hydrostatic equilibrium (HE) to convert ice elevation change observations to thickness changes. Additional justification of this approach is needed, along with a more open discussion of the associated limitations and uncertainties in the main text.

While using HE for this calculating ice thickness is appropriate for fully floating ice, the InSAR data indicate that the ice in the Berry IGZ is grounded for a significant portion of the tidal cycle, with an average depth of seawater intrusions of ~ 0.6 m. This implies the ice within the IGZ cannot be in hydrostatic equilibrium, casting doubt on the validity of ice thickness values derived from this method, and, by extension, the estimates of thinning rates, ice discharge, basal melt rates and bathymetry. For example, the authors' interpretation that a 24 m surface elevation decrease in the IGZ translates to 222 m of thinning (L101-102) is likely to be a significant overestimation.

It is well-documented that satellite-derived estimates of ice thickness and basal melt rate within GZs are subject to large uncertainties due to the breakdown of the hydrostatic assumption (e.g., Chuter & Bamber, 2015; Griggs & Bamber, 2011; Kim et al., 2024), which often leads to these regions being masked out (e.g., Griggs & Bamber, 2011; Paolo et al., 2015; Wilson et al., 2017) or interpolated across (e.g., Fretwell et al., 2013; Morlighem et al., 2020). While I note that some attempt has been made by the authors to address this (e.g. L150-152; 296-298), the discussion remains too vague and does not adequately address the impact on the key findings.

Response: As ice is afloat during part of the tidal cycle and the tidal amplitude is only ± 0.6 m, ice must be within 1-2 meter of flotation, hence the error of using HE during the tidal cycle is less than 10-20 m. Following the 24 m surface lowering (Line 108), the grounding line reached state 2 in 2006, so the assumption of HE is valid. Finally, in Figure 3g, we match the bed depth from MCoRDS with the ice draft derived from the 2019 DEM, which gives another line of evidence that this calculation is reliable. This is now stated more clearly in the manuscript.

3) Refinement of bed topography

The authors propose that the bed topography within the Berry IGZ is up to 1,300 m deeper than measurements from MCoRDS airborne radar and in BedMachine v3.7 (which interpolates between sparse radar measurements). However, I do not think that the evidence presented here is strong enough to support this conclusion; I suggest the authors either reconsider this interpretation or provide a more robust justification, clearly communicating

any limitations. Specific concerns include:

a) Methods description: The description of methods used to reconstruct bed topography lacks sufficient detail, both in the main text (L125-127) and methods (L278-279). I recommend expanding the description to clarify the approach, including details on the geoid and specifying the areas or flowlines over which bed topography was calculated.

b) Use of HE/flotation: If the new bathymetry estimates in the IGZ are based on ice thickness calculated using HE/flotation, they are subject to the same uncertainties outlined above. Prior studies have demonstrated that ice thicknesses calculated in this way close to the GL tend to overestimate values compared to radar measurements due to the breakdown of HE (e.g. Chartrand & Howat, 2023).

c) Radargram interpretation: The authors propose that the bed depths from the MCoRDS radar in the Berry IGZ have been misinterpreted as too shallow (L132-134). However, upon reviewing Supp. Figure 2 (noting that I am not a radar specialist), I find this interpretation unconvincing. While weak bed echoes at the glacier center may well indicate higher uncertainty in bed depths, the picked bed depths (red dashed lines) in all four radargrams look reasonable and there is little indication of any deeper echoes at the depths suggested by the authors. Radar is generally considered most reliable way to measure ice thickness/bed depth in these regions, and it seems more plausible that the HE-based calculations in this study might overestimate thickness/depth compared to the radar. Labelling specific features in the radargrams that support the deeper bed interpretation would be helpful, as well as providing a comparison to data from the gravity inversion mentioned in L136.

d) Clarification of the 1300m depth difference: The origin of the 1,300 m value is unclear. In Figure 3(g-h), the difference between the BedMachine bed elevation and the author's new calculations appear to max out at around 1,000 m. Since this 1,300 m value is presented as a key result in the abstract, its basis should be clarified in the text.

Response: 1. As stated in Lines 348-349, the EIGEN-6C4 geoid model from BedMachine v3.7 is used to geoid-correct the DEMs. The spatial extent over which the bed topography is reconstructed is clarified in Lines 137-139 and 340-341; 2. As noted earlier, the agreement between the ice draft inferred from HE and the basal interface detected by MCoRDS radar supports HE in this region; 3. The MCoRDS data document the ice-bed interface; therefore, in the IGZ, the bed elevation inferred from MCoRDS cannot be below the HE-inferred ice draft. Along profile A-B, the radar signal over the central IGZ region (above the red dashed line) suggests that the bed was misidentified, prompting us to shade the area between the MCoRDS-derived bed and the ice draft to highlight this uncertainty. Along profiles C-D and C-F, radar wave reflections from the steep valley sidewalls (off-nadir returns) obscure the basal interface in the valley center. For profile E-F, discrepancies are observed between the 2011 OIB Level 2 dataset and the corresponding radargrams: while the radargrams have the correct bed picks, the Level 2 data contained erroneous values, leading to the substantial mismatch between our reconstructed bed topography and BedMachine. Unfortunately, users have used the Level 2 values instead of the Level 1 values. We have updated Supplementary Fig. 2d to show the erroneous data for clarity; 4. In the revision, we note that the maximum depth discrepancy of 1300 m is located on the eastern side of the IGZ, but it does not overlap with any radar profile examined in this study (See Lines 140-143), so it is a maximum offset with BedMachine which is interpolated data at this location.

4) Long-term melt rates / basal melt regime

The text in this section (L146-192) needs refining; it is currently difficult to follow the specific areas and time periods for which melt rates have been calculated. Clear distinctions are needed between discussions of: (a) Melt rates within the Berry IGZ (2019-2022); (b) Melt rates within a zone 10 km seaward of the IGZ, compared to neighboring glaciers and previous measurements; and (c) Predicted melt rates within the Berry IGZ due to tidal seawater intrusions. Again, my concerns about the use of HE to calculate thinning rates within the IGZ alongside the lack of uncertainty quantification make me skeptical about

the validity of the derived melt rates. The authors briefly mention using 4.5 km spatial smoothing to mitigate deviations from hydrostatic equilibrium (L150-152), but further detail is needed to explain what was done and how this reduces uncertainty. As I understand, the authors have interpreted the formation of a >100m thick cavity beneath the Berry IGZ over the past 25 years (L156), reaching depths up to 300 m near State 2. This implies the whole region is afloat, which starkly contradicts the InSAR GL results which show the whole IGZ is ephemerally grounded with depths of seawater intrusions averaging just 0.6 m. Clarification of this discrepancy is important.

Response: 1. The distinction of areas and periods has been updated at Lines 162-163 and 190-193; 2. We used different window sizes for the filtering until the melt rates do not change at a detectable level and heterogeneities are eliminated. The window size is 4.5 km, which roughly corresponds to the size of the bending zone discussed by Chartran et al. 2024; 3. We attribute the formation of the subglacial cavity at state 2 to basal melting. The DInSAR fringe patterns observed in this region reflect the vertical displacement of the ice surface driven by subglacial water motion, rather than the total water thickness. In other words, the cavity has areas of partial grounding, and the water column must be thicker in between these pinning points, perhaps 10's m, but we do not have data to quantify it.

5) Tidal seawater intrusions

The estimates of melt rates due to tidal seawater intrusions (L181-192) appear to rely heavily on assumptions and values that are either poorly constrained or lack clarity on their uncertainties. This undermines confidence in the results and may mislead readers, especially if the values are subsequently used for in models. It is recommended that the authors clearly outline all assumptions, cite where the values are taken from (note specific comments below about the CTD data), and acknowledge the scale of the limitations.

Response: Yes, the melt rates quoted in this section are order-of-magnitude melt numbers. In the revision, we made this clearer (Lines 205-225).

4. Diversity, Equity and Inclusion: I would like to draw attention to the lack of female representation among the nine co-authors, which might be something the authors or the journal could consider addressing in future to foster more diversity in scientific collaborations.

Response: This comment is not appropriate in a scientific review.

5. Specific Comments

- 1) Title – The title doesn't really reflect the key findings of the paper; I suggest this could be updated to include something on seawater intrusions which I think are arguably the most interesting result. Also 'and other data' is a bit weak.

Response: The title has been changed to "Rapid retreat of Berry Glacier, West Antarctica, linked to seawater intrusions revealed by satellite radar interferometry data".

- 2) Abstract – should be updated to reflect any changes made to the manuscript during revision.

Response: The changes in the manuscript are updated in the abstract.

- 3) L42 – typo; 'affected'

Response: Fixed at Line 43.

- 4) L47-49 – These sentences are quite vague. Are you referring to the effect of short-term GL retreat and advance periods (i.e. with the tides), or longer-term retreat/advance phases? Please clarify and consider adding references for support.

Response: If we refer to "GL retreat reduces bottom friction and speeds up the glacier, GL advance increases drag and slows down the glacier", then it is the same effect on short and long time scales.

- 5) L51 – This paper doesn't directly discuss CDW; you might consider citing more relevant oceanography papers, such as Dinniman et al. (2012) <https://doi.org/10.1175/JCLI-D-11-00307.1>, Wåhlin et al. (2010) <https://doi.org/10.1175/2010JPO4431.1>; Schmidt et al. (2023) <https://doi.org/10.1038/s41586-022-05691-0>; Thoma et al. (2008) <https://doi.org/10.1029/2008GL034939>.

Response: We added these references at Line 72: Thoma et al. 2008, Dinniman et al. 2012, and Schmidt et al. 2023.

- 6) L66 – I recommend using the full name 'Berry Glacier' here, since this is its first mention in the main text.

Response: Added at Line 64.

- 7) L67 – 'locus' suggests a single point; consider using 'area/region'.

Response: "locus" changed to "region" at Line 65.

- 8) L75 – The dash between 0.6 and km can be removed here, and similarly in other places where it is not used as an adjective.

Response: The dash has been removed at Line 75, and other places.

- 9) L79 – It is not clear where this observation is shown on Figure 2; I suggest adding lines/labels

Response: The GL from HE has been added to Figure 2a.

- 10) L87-98 – This analysis would be strengthened by discussing potential reasons for the lack of correlation between tidal forcing and short-term GL migration, as well as considering other controls that may drive this short-term variability. Comparing to studies that have documented different patterns of short-term GL migration elsewhere in Antarctica (e.g. Rignot et al., 2024; Chen et al., 2023; Freer et al., 2023) would be valuable. For example, could this interpretation be affected by calculating modelled tides 50-100 km seaward of the IGZ, where tidal forcing may be out of phase with the resulting tidal flexure within the IGZ itself? Exploring whether there any correlations with the longer-period spring-neap tide cycle, phase lags between rising and falling tides, or interactions with upstream subglacial hydrology—as observed in the Thwaites IGZ (e.g. Chartrand et al., 2024; Rignot et al., 2024)—would enhance the analysis. I note that Figure 5b shows several subglacial channels upstream of the IGZ, but this has not been addressed in the text.

Response: The spreadsheet in the Dryad archive includes tidal height computed at two offshore locations, 50 km and 100 km seaward of the IGZ. The record shows negligible difference in tidal amplitude and phase between the two sites. To further investigate the tidal influence on GL migration, we analyzed the spring-neap tidal cycle as well as the ebb and flow phases for each GL state, as now shown in Supplementary Fig. 1. Our results show no discernible correlation between grounding line retreat and sea surface height, even when accounting for variations caused by the spring-neap cycle and tidal phase.

- 11) L87-90 – It would also be interesting to see a comparison of the modelled Δ tide (at this same position 50-100 km seaward of the GL) compared to the measured InSAR Δ SSH within the IGZ. This may provide some insight into the non-linear relationship between tidal forcing further from the ice front and the ice shelf flexural response/ GL migration within the IGZ.

Response: This was shown in Figure 2c, which indicates that tide model and DInSAR agree.

- 12) L97 – I suggest it is challenging to confidently state that the GL was "predominantly" in State 1 in 1996 based on only two observations. Perhaps rephrasing to reflect this uncertainty would be provide more context.

Response: Yes, we added a note of caution and indicated no evidence for signal upstream of that position in 1996, in contrast with recent interferograms (See Lines 76-80).

- 13) L101-115 – This section is somewhat challenging to follow as currently structured. Consider reorganizing and re-wording to improve clarity. It appears to mix different time periods (2006-2022 and 1996-2022), various regions (e.g., "between states 1 and 2," "west flank," "between the 1996 and 2008 GLs," "from the 2008 GL to 28 km beyond the 2021 IGZ upper limit," "along profiles G-H and C-D," "glacier on average"), and different values (ice elevation, thickness, thinning, speed, and speed-up). This makes the text difficult to follow. Additionally, when Δh , ΔH and ΔV are averaged over specific areas, these areas should be clearly defined or located on a map.

Response: Agreed. We revised the paragraph for improved clarity and logical flow (Lines 106-123). We clarified our description of the western flank (Lines 110-111), now explicitly defining it as the region where the IGZ expanded laterally. Figure 3b highlights the area of ice flow acceleration, extending up to 28 km beyond the 2021 inland limit of the IGZ.

- 14) L104 – Clarification is needed regarding how glacier speed changes were calculated; not enough detail is given here or in the methods section. Are these glacier speed changes calculated as an average along flowlines, at specific points, or averaged across specific areas? Including this information in the text or figure would provide useful context.

Response: The reported speed changes represent average values calculate over specified regions or along specific profiles. We clarified each case in the revised text.

- 15) L105 – The sentence on glacier speed feels out of place between other sentences discussing glacier speed-up. Consider moving it to a more logical position or removing altogether. It is important here to be specific about the time stamp of the velocity value.

Response: The purpose of referencing the maximum speed here, along with the subsequent statement, was to highlight that the location of peak ice speed does not coincide with the region exhibiting the largest speed change. The sentence has been removed for clarity.

- 16) L106 – The 1996 and 2008 GLs are not currently shown on Supplementary Figure 1, but should be added.

Response: Based on the comment above, we removed the sentence related to the maximum speed, the only place where Supplementary Fig. 1 was cited, therefore we removed the Supplementary Fig. 1 as well.

- 17) L107-109 – It is important to clearly define the region over which the average speed-up of 134 m/yr and a 59 m elevation decrease is calculated.

Response: Agreed. As noted at Line 115-117, the region refers to the area extending from the state 2 to inland by 28 km beyond its upper limit of 2021 IGZ.

- 18) L110-111 – The results in L100-109 discussed change in V and h over the period 2006-2022, but it now seems for the flowline results you switch here a longer period 1996-2022. Please explain this discrepancy in time periods, and ensure the text explicitly mentions these timeframes to avoid confusion.

Response: Agreed. In the revision, we separate the discussion of the change in 2006-2022, for which we have complete spatial maps for the basin, from the discussion for 1996-2022 for which the 1996 have data gaps, therefore we only analyze profiles.

- 19) L110 – Consider rephrasing for clarity: ‘... the surface elevation decreased by up to 59 m and ice sped up by 48% from 448 m/yr in [YEAR] to 665 m/yr in [YEAR]’.

Response: The sentence has been updated at Lines 118-119.

- 20) L113 – is ‘on average’ referring to the entire glacier basin?

Response: The statement refers to the 2021 IGZ. See update at Line 121.

- 21) L113 – As explained in the general comments above, I suspect these thinning rates are significantly overestimated given the region is not in hydrostatic equilibrium (especially if these values are referring to the entire glacier basin, the majority of which is fully grounded).

Response: We apply the hydrostatic equilibrium (HE) assumption only to regions where seawater intrusions are inferred at a specific time period. For the period 1996–2005, changes in ice thickness were equated directly with surface elevation changes, as the GL reached to state 2 around 2006. Between 2006 and 2011, the HE assumption was applied within the region from state 1 to state 2. From 2012 to 2022, HE was applied across the entire 2021 IGZ extent.

- 22) L114-15 – Is this referring to the spatial pattern of thinning across the whole basin? I suggest you are more explicit about what is being cited here from Selley et al. (2021); is this an observation they also made?

Response: This was a mistake. Text has been corrected.

- 23) L116-123 – I recommend the authors clarify the time period and locations for ice discharge calculations, and how this relates to the two time periods of observed h and V change discussed above (2006-2022 vs 1996-2022). It is currently difficult to ascertain how this was done.

Response: The time period and locations of flux gates for ice discharge are indicated in Fig. 1a and mentioned more clearly at Lines 316-320. The relationship between ice discharge and changes of h/V is discussed at Lines 327-329. We initiate the ice discharge calculation in 1996. For the time periods 1996-2005, 2007, and 2013, a linear interpolation is used to fill gaps in ice discharge as mentioned in the caption of Fig. 4. This is now clarified at Lines 124-126 and 316-331. The time period and locations for the ice discharge calculations are related to the timing when the GL reaches each state.

- 24) L116 – Against which baseline are these discharge anomalies measured? Is it the same 1979-2004 reference period cited for the SMB anomalies in the following sentence? If so, please clarify.

Response: The reference SMB is 1979-2004 is used as the balance discharge. See Lines 126-130 and 334-337.

- 25) L119 – Consider rewording for clarity: ‘Between 1996 and 2022, annual anomalies in mass balance, M, increased by a factor of 6.6, from X Gt/yr to Y Gt/yr’.

Response: The sentence has been re-written at Lines 131-133.

- 26) L122 – It would be more logical to introduce the total increase in ice discharge at the start of this paragraph, and then transition from discharge > SMB > mass balance.

Response: The sentence has been rewritten at Lines 133-135.

- 27) L127 – ‘jack up’ is a somewhat colloquial phrase which can have negative connotations; a different term like ‘lift up’ would be more suitable here.

Response: Agreed. The sentence has been rewritten at Lines 137-139.

- 28) L131 – The interchangeable use of "OIB" and "MCoRDS" in the text can be confusing. Consider specifying "MCoRDS" when referring to radar-derived thickness data and "ATM" for altimetry data from OIB and updating this consistently throughout.

Response: Agreed. “OIB” has been replaced by “MCoRDS” when referring to radar-derived thickness.

- 29) L141 – It would help to clarify here that you are comparing results to a modeling study, not direct observations. Consider rewording to something like: "... gravity gradients,

as modeled in [20].".

Response: The sentence has been rewritten at Line 156.

- 30) L144 – A number of previous studies have also observed this; you may wish to cite works such as Chen et al. (2023), Freer et al (2023), Rignot et al. (2024).

Response: These citations have been added at Line 160.

- 31) L146 – Please specify the time period over which basal melt rates were calculated—was this solely for 2019-2022, as indicated in Figure 5?

Response: The time information has been added at Line 163.

- 32) L154 – It would be helpful to clarify which area is being referred to here and to mark this region on Figure 5.

Response: This has been clarified at Line 168. The region is the IGZ.

- 33) L156 – you should use “meters” instead of “m’s”.

Response: “m’s” has been changed to “meters” at Line 172.

- 34) L171-173 - There appears to be a contradiction between these sentences. The text initially states that the Berry IGZ was unknown during previous studies, but then references comparisons of melt rates "in the IGZ of Berry" between 2003-2008 and 2019-2022. Please clarify this.

Response: Yes, we needed to clarify that we compare the melt rates within 10 km of the lower limit of the IGZ as stated at Line 190 where we have coincident measurements.

- 35) L172 - The “area of coincident measurements” for comparing the three time periods of basal melt rates is unclear. I suggest the authors label this area in Supplementary Figure 5. This applies to the areas used for the melt rate calculations at the other glaciers, such as De Vicq, as well.

Response: We added white dashed polygons in Fig. 5c–d and Supplementary Fig. 5 to delineate the region of coincident measurements.

- 36) L176-177 – The statement "the melt rates in the IGZ of Berry are also the largest for the Getz Ice Shelf" appears in a section discussing melt rates in the 10 km downstream of the IGZ, rather than the IGZ itself. Consider moving this sentence to the previous section or clarifying that it refers to the downstream zone.

Response: “the downstream” has been added at Line 201.

- 37) L177-180 – I suggest adding a reference to Supp. Fig 5d here, as that is the only place this result is shown.

Response: The reference has been added at Line 204.

- 38) L182 – There should be a unit given in ‘... from 0 m at state 3 to...’.

Response: The sentence has been re-phased at Lines 205-206.

- 39) L185 – Does the 0.6 m value represent a calculated average of the observed InSAR displacements? Or is this a prediction? More clarity (+associated uncertainties) is needed.

Response: “0.6” is half of the maximum vertical amplitude at state 1, 1.2 m, which comes from the tidal model. “Observed vertical displacements at the entry of the IGZ of 1, 4 and 4.7 CSK fringes translate into a minimum change in water column of 1.4, 5.5 and 6.5 cm, respectively” has been deleted to avoid confusion.

- 40) L186 – the stated water flux of 6.9×10^7 m³ in 6 hours would assume that the GL migrates all the way to state 3 each tide cycle. However, your results indicate there is

no observed correlation between tide forcing and GL migration distance (Fig 2) – so do you think this is still a reliable estimate of water flux? Please clarify and/or be more open about the associated limitations of this assumption.

Response: This calculation is a maximum melt rate if water intrusions flush the entire cavity and all available heat is used to melt the ice.

41) L186 – You use an area of 114 km² for the IGZ here, but earlier a 100 km² IGZ area was used to calculate melt rates (L154)? Clarifying why these values differ would be helpful.

Response: Agreed. “100 km²” has been changed to “114 km²” at Line 168.

42) L187 – This is the first and only time you mention a 2018 CTD, but there is not associated citation. Earlier, you referenced a 0.7C ocean temperature at 1,000m depth in front of Berry measured from a 2007 CTD (Jacobs et al. 2013). The ‘2018 CTD’ data gives ocean thermal forcing at 1,000m depth of +3.6C. Is this a mistake in the text with the date, or are you referencing two different datasets here? If this is the case, please clarify the date and location, alongside the appropriate citation.

Response: We originally cited Selley et al. 2021. Upon careful examination of the CTD dataset from KOPRI—used in the 2021 study—we found that the measurements extend only as far west as Dean Island but do not reach Berry Glacier. Consequently, to estimate post-2007 ocean temperature changes at the ice front of Berry Glacier, we analyzed CTD profiles collected at distant locations in west Getz Ice Shelf after 2007 instead (Supplementary Fig. 6). The CTD records indicate a warming trend since 2000, which persisted beyond 2007. If we assume that the ocean in front of Berry Glacier experienced a comparable degree of warming, we find a thermal forcing of +3.6°C at 1,000 m depth in 2018. This discussion has been added at Lines 208-216.

43) L188-189 – I suggest more detail is needed regarding the calculation of ocean heat and freshwater fluxes, including any associated uncertainties or assumptions.

Response: The primary sources of uncertainty in the ocean heat estimate stem from two components: the thermal forcing and the volume of intruding seawater. We revised the calculation using the CTD from 2018 (Supplementary Fig. 6). For the intrusion water volume, we assume an average cavity thickness of 0.6 m. We estimate a low efficiency of the melt process at 30% and obtain an average melt rate of 6 m/year which indicates that water thickness in the IGZ must be higher than 0.6 m as noted at Lines 221-225.

44) L190 – the statement that ‘only 30% of the heat will be used to melt ice’ is based on modelling at Petermann Glacier in Greenland. How confident are you that this can also be applied in this setting? Without providing uncertainty estimates or being open about these limitations, this quantification could be misleading.

Response: The citation of the $30 \pm 10\%$ is from a simulation of ocean circulation using the MITgcm ocean model in a grounding zone. There is no fundamental reason for the physics to be different in Greenland vs Antarctica. We revised the calculation, fixing the error in water height, and used a more recent value of ocean temperature from 2018 at Lines 205-221.

45) L191 – Please clarify the basis for the statement that the seawater intrusions may be “thicker by a factor of two”; has this been quantified or just speculative?

Response: Sentence has been removed.

46) L206 – In general, the methods lack sufficient detail to reproduce the analysis, although perhaps this has been limited by the word count of the article. Citations should be provided for all datasets.

Response: Reference has been added.

47) L218 – I suggest this should cite a paper describing the Sentinel-1 instrument, rather than an ice sheet study that has also used Sentinel-1.

Response: The citation of the Sentinel-1 instrument has been added at Line 264.

48) L222-228 – This section is missing a description of how the differential SSH was calculated from the interferograms, and whether the GLs were delineated manually or using an automated approach. Also, how do you define a ‘high quality’ interferogram?

Response: The descriptions of how the differential SSH is calculated and the method for delineating the GLs have been added at Lines 283-283 and 273, respectively. The “high quality” means we have a high-precision, low-noise determination of the GL position.

49) L226 – should be ‘dense’ instead of ‘densely’.

Response: Changed at Line 272.

50) L226 - The attached data spreadsheet includes a column for GL migration distance ('GL migration_Berry') as well as columns of 'positive error' and 'negative error', however there is no description of these errors in the text. I suggest the authors include a robust description of these error terms and the calculations used to derive them here.

Response: When the fringes are noisy or merge or split, we estimate the positive and negative errors based on the fringe pattern to show the potential minimum and maximum position of the GL. See Line 274-276.

51) L249-258 – The temporal coverage of each of the elevation change products used here should be provided:

- ICESat-2 ATL14 DEM is timestamped to Jan 1 2020
- ICESat-2 ATL15 elevation change products – between 2019 and 2024
- Add date ranges for other satellite missions used in the MEaSURES elevation change products.

Please clarify whether you use the ATL15 ICESat-2 elevation products (1km) as well as the MEaSURES product using ICESat-2 (2 km)? Or just the former? And this section is missing information and citation for the WorldView-derived DEM and ATM data (e.g. used in Fig. 3, and in calculation of ice bottom and bed topography below).

Response: The information of the products above has been added in Lines 304-313. We only use the MEaSURES product for the period of 1996-2018. We do not use ICESat-2 at 2 km resolution. The citation and information for WV and ATM have been added in Lines 343-345 and 321-325.

52) L262-3 – this seems to be a new result that hasn't been presented in the main text? Please clarify.

Response: Agreed. The elevation change ($\partial h/\partial t$) along flowlines, derived from ICESat-2 ATL15 and MEaSURES products, was not intended to be a new, independent result. Rather, it serves as supporting evidence to corroborate the GL migration established based on DInSAR-derived GL positions. These indicators help understand the GL migration and the placement of flux gates at different times.

53) L278 – The current description lacks sufficient detail. In the main text it says the bed topography is reconstructed ‘based on the minimum depth of ice that ensures flotation of ice over the zone of flexing, i.e. where a meter of change in SSH is sufficient to jack up the ice surface’. Please elaborate here on this calculation, specifying e.g. the geoid used and whether the reconstruction was performed over the whole IGZ region, or along individual flowlines?

Response: Agreed. We elaborated on the bed reconstruction method in the revised text. Specifically, the reconstruction was performed across the entire IGZ, see Lines 340-341. The method involves calculating the minimum ice draft required to ensure flotation based on the surface elevation data (using the BedMachine v3.7 geoid; Lines 348-349) and firm

air content. After corrections, we calculate the minimum depth of the ice bottom (Lines 340-353).

- 54) L281 - Given that FAC values are known to be uncertain over GZs, it would be important to acknowledge this uncertainty here (even if it cannot be precisely quantified). Additionally, consider citing the firm model used in BM3.7 to provide more context.

Response: Agreed. The citation about the firm model have been added at Line 349-350. Although the uncertainty in firm air content (FAC) over grounding zones is not quantified in BedMachine v3.7, Ligtenberg et al. (2011)—the basis of the firm densification model—highlight that firm properties exhibit substantial spatial variability across Antarctica, and that firm correction values may vary up to 7–9% in coastal and shelf regions. The authors acknowledge that their model is semi-empirical and does not fully capture the physical complexity of firm processes in dynamically evolving zones such as the IGZ where complex processes such as surface melt, tidal flexure, and dynamic thinning may deviate from the modeled steady-state assumptions. We added a statement to the revised text at Line 350-353.

- 55) L283-286 – see previous comment on this interpretation; I would suggest it's more a case that your calculations are inconsistent with the radar measurements (rather than the other way round). A stronger justification is required, beyond simply that than the L2 data is 'not as robust'.

Response: The GL positions derived from DInSAR suggest that portions of profiles A–B, C–D, and C–F are partially afloat. Under these conditions, the bed elevation inferred from MCoRDS radar sounding cannot be deeper than the ice draft derived from HE using ATM surface elevation data. In our previous response, we discussed in detail the limitations of the MCoRDS-derived bed along these profiles, including misidentification of the basal reflector due to off-nadir returns or weak radar backscatter in the valley center. These issues are well documented in the radargrams and explain why the radar-derived bed is inconsistent with flotation-based estimates. It is not the first time erroneous bed picks have been reported from MCoRDS.

- 56) L289 – It would be useful to see a justification of applying a Eulerian vs Lagrangian approach here, or acknowledgement of the associated uncertainties when applied to floating ice.

Response: Agreed. We clarified in the revised text that the use of a Eulerian approach is required for ICESat-2 ATL15 data. Specifically, ATL15 provides surface elevation changes relative to a fixed DEM, rather than absolute surface elevations at discrete time steps referenced to moving ice parcels. We acknowledge that applying the Eulerian method to floating ice introduces additional uncertainties, particularly in regions experiencing advection or basal geometry changes. To mitigate these effects, we have applied a spatial filter based on the length of the bending stresses and the size of heterogeneities. We included this discussion in the revised Methods section (Lines 372-378).

- 57) L293-4 – Please provide more detail on this, as per previous comment.

Response: The information regarding uncertainties has been provided at Lines 383-385.

- 58) L295-299 - It is good that the potential overestimation of melt rates is acknowledged here. However, this point should be emphasized more strongly within the main text, accompanied by a more detailed explanation of the uncertainty estimation process. Specifically, it is unclear how you arrived at the stated uncertainty of "about 36%". It would be helpful to clarify this methodology, and any reported uncertainty values should be presented alongside the corresponding results in the main text for transparency and clarity.

Response: We evaluate the uncertainty at 21%, see methods section, where we explain the

details.

6. Figures

- 1) In general, the figures and captions (in main text and supplementary) deserve some attention to bring up to publication standard. Most text is too small to read comfortably, there are a mix of font types and sizes, and overlapping data and sub-panels obscures some important information. I recommend using a consistent structure for figure captions throughout, i.e. Fig X - main title. (a) subpanel caption; (b) subpanel caption; (c) etc. I have made some suggestions here for a number of (relatively minor) edits that could be made to the main text figures that I believe would significantly improve their quality. The same principles should be applied to the supplementary figures.

Response: Thank you. We increased the size of letters and made the styles of the figures more consistent.

- 2) Figure 1: This figure contains a lot of valuable and important data, but its impact is currently limited by the way it is presented. I suggest that the authors should consider re-formatting to a portrait layout, which would allow it to be displayed as a full-page figure. Alternatively, consider separating panel (a) as its own figure, and organizing panels (b-g) into a separate portrait figure (e.g. a 2x3 grid). Some suggested improvements include:
 - Panel (a) –
 - o Avoid use of $\langle \rangle$ symbols in the velocity legend for clarity.
 - o The colors of the flux gates get a bit lost; perhaps keeping these in bold black while reducing the opacity of the rest of the OIB lines would improve clarity.
 - o Add a box to the map to show the area in panels (b-d).
 - o The elevation contour values could be removed; they are somewhat redundant alongside the blue elevation color scale. This would simplify things and avoid contour values overlapping the main data you are presenting (i.e. the GL positions).
 - Panels (b-d)
 - o These panels are well-presented, but the caption should provide more information (i.e. explain these are interferograms composed from images sampled from the 3-4 tidal states shown in panels (e-g); and describe what the color bar represents).
 - o Adding labels for the maximum tide height and/or differential SSH of each interferogram could assist readers in understanding the data presented.
 - o Clarify whether the inclusion of three interferograms comprised of scenes all sampled at low tides is a deliberate choice; if so, consider briefly explaining this in the caption as it is interesting for the interpretation!
 - Panels (e-g)
 - o Consider reducing the number of values on each axis label (e.g. intervals of 12h on x-axis) to allow for an increase in text size.
 - o The grey lines pointing to sections of the longer tide time series clutter the figure. Consider removing or ensuring they do not cross over one other.
 - o Consider repositioning the inset panels to prevent them from covering important data on the main panel, as this is an issue across all three panels (e-g) where some of the inset panels cover the red dots.

Response: Figure 1 has been updated based on the comments, along with the caption. The interferograms are not chosen deliberately for the low tides. We choose them to show the best example of each state.

3) Figure 2:

- Panel (a)

- o This density plot is a very effective way to illustrate the three GL states.
- o Some clarification is needed for the values presented in the DInSAR/HE box. Consider adding details to the caption, including an explanation of HE as hydrostatic equilibrium, and incorporating lines on the map to show the locations of the HE GLs for better reference.

- Panel (b)

- o Is there a particular reason for using a different color map and phase difference scale compared to Figure 1(b-d)? I'd say the color scale in Figure 1(b-d) is clearer.

- o From your caption, I would expect the downstream seawater intrusions to be marked by a region of uplift, and upstream extrusion of seawater trapped in a prior cycle marked by a region of subsidence – but it appears that these labels are reversed on the map? Or is the region of extrusion further upstream (between states 2 and 3)? This may be my own misinterpretation, but it would be good to clarify this. Adding labels to show the regions of intrusion vs extrusion would be beneficial.

- o Including arrows above/below the legend to illustrate the meaning of the diverging fringe colors could greatly enhance comprehension of this figure for those less familiar with DInSAR maps (i.e. pink>yellow>blue = uplift; and blue>yellow>pink = subsidence).

- Panel (d)

- o This is an effective presentation of these results. A few minor clarifications would further aid interpretation:

- Indicate what the error bars represent (both for modelled max tide and GL migration distance).

- Confirm that h_{\max} refers to the maximum modelled tide and clarify in the caption.

Response: Figure 2 has been updated based on the comments. The caption related to intrusion/extrusion is corrected. 1. The different color map and phase difference scale for the interferograms are because the CSK and Sentinel-1 are generated with different scaling; 2. For panel (d), the horizontal error comes from tides model and mean sea level pressure, and the vertical error shows the range that the farthest and closest distance that the GL may reach because when the fringe boundary is unclear or fringes merge or split.

4) Figure 3:

- The caption currently lacks clarity, specifically,

- o L2 – Clarify that (a) and (b) represent change in h / change in V

- o L3-4 - Remove 'Year 2006 is the reference for (a-b)'; this info is already conveyed in the preceding text and figure.

- o L4-5 – Maintain consistent phrasing for panels (c,d) and (e,f), i.e. 'V(/h) along profiles G-H and C-D for each year 1996-2022'

- o L6 – specify that the elevation data is from the WorldView (WV) DEM (i.e. not just from WV imagery)

- o L6-12 – separate captions for panels (g) and (h); currently confusing as different datasets are used in each for the same thing.

- o L6-12 – Clarify descriptions of the different blue lines ('blue line'/'blue dash line'/'blue solid line'/'dark blue line'). It is confusing where the same line style is used in panels (g) and (h) to show different data (e.g. in (g) the solid light blue line is from the 2019 WV DEM, but in (g) the same line style shows the '2012 DEM'). Consider changing the line styles or providing a more explicit legend.

- o L7 – please clarify which DEM is used for ice draft in (h)? It is unclear what instrument is used for the '2012 DEM' – it says ICESat-2 in the caption, but this cannot be correct as ICESat-2 only launched in 2018.

- o L8 – please clarify if by 'flotation from ICESat-2' you mean: flotation depth derived from the 2020 ICESat-2 DEM?

- o L10-12 – ensure consistency in the description of the OIB data points.

- GL positions/IGZ zones
 - o On panels c, e and g, it is confusing having both the horizontal and vertical bars showing GL locations, positioned both above and below the main data. I suggest keeping just the horizontal bars for 2019-21 to show the short-term range of GL positions in each of the 'states' and labelling the three states in each panel (as has been done in g). Then you can use a vertical bar/dot for 1996 and 2008/9 GL positions where you don't have sampling of the IGZ length.
 - o For panels d, f and h, it is fine to use vertical lines, as this is showing the total width of the glacier/shelf.
 - o I suggest positioning all GL positions at the bottom of each panel (including in g) for consistency, rather than having some above and below the main data.
 - o Reflect any changes made in the caption.
- Panels g-h – These are currently quite confusing to interpret.
 - o Consider moving both the x-axis and IGZ labels to the bottom of each figure (rather X than in the middle of the glacier).
 - o Clarify if elevation measurements are relative to mean sea level
 - o Legend and caption currently missing info on the ?s, striped areas and blue shaded areas.
 - o It is currently unclear what the areas labelled as ice/bed/water are using data from? BedMachine? Or the bed you've deduced from flotation? In (h) looks like the 'bed' is from the OIB data, but in (g) it's from a mixture of BedMachine and your WV-DEMderived bed.
 - o In panel (g), the blue dots representing the OIB data are not easily visible. The legend entry for OIB reads like a date (i.e. 2011/12/16 could be interpreted as 16 December 2011). I suggest since there are only four OIB points on this panel, each could be individually labelled with their measurement year.
 - o There doesn't seem to be a consistency in the results at the expected crossover point between profiles G-H in panel (g) and C-D in panel (h); perhaps this is the result of using the same colors in both panels for different data?

Response: Figure 3 and its caption have been revised in response to the reviewer's comments. 1. Elevation measurements are referenced to mean sea level using the EIGEN-6C4 geoid model from BedMachine v3.7 for geoid correction; 2. Descriptions of the shaded areas have been added to the figure caption and the main text (Lines 359-365); 3. We clarified the priority order for bed topography data sources: where available, we adopt the bed from MCoRDS; within the IGZ, we use flotation depths derived from geoid-corrected surface DEMs when MCoRDS is not available, as BedMachine performs poorly over Berry Glacier; outside the IGZ, BedMachine is used; 4. The crossover point between profiles G-H and C-D in panel (h) does not correspond to the deepest part of the bed. We have added the crossover locations to the X-axes in the relevant panels for clarity.

5) Figure 5:

- The sizing of this figure is much better
- On panel (b) red GLs get lost in the red melt areas; consider using a different color scale?

Response: A thin white outline has been added to the red GLs to enhance their visual contrast against the background in Figure 5.

7. Datasets

The authors have provided easily accessible datasets of derived GL positions, flux gates etc. that were straightforward to open in GIS software. I recommend that all processed interferograms should also be included in the dataset. The excel spreadsheet 'GL analysis, tidal heights, speed & surface height.xlsx' contains a lot of useful data, but some clarification and reorganization is required to improve its usability. Notably:

- Columns F and G ('positive error' and 'negative error') appear to indicate the number of days between the first and last SAR image date and the midpoint between these dates. Please clarify how this constitutes an error, and—if so—how this error is dealt with in the subsequent processing.

- Columns P and Q are also labelled 'positive error' and 'negative error', but these appear to indicate some error in the derived GL position. Please provide an explanation of this error and its calculation, assumptions etc.

- This dataset contains columns for modelled tides/IBE at 50 km and 100 km. Please clarify what these distances refer to – i.e. is it the distance between the downstream IGZ limit and the position used to calculate the tide? If so, which of these locations (50 or 100 km) did you use? If only one location is used for the tide calculation, only these values should be included in the dataset to avoid confusion.

- It is confusing that the SAR image acquisition dates are repeated three times in the same Excel sheet: in columns A-D, S-V and AP-AS. There is also repeated GL migration data in columns O ('GL migration_Berry') and AM/AN ('GL migration_Sentinel-1'/'GL migration_COSMO-SkyMed'). Consider consolidating these columns to remove unnecessary duplication.

Response: 1. The interferograms generated from Sentinel-1 and CSK cannot be uploaded due to copyright; 2. The purpose of the "positive/negative error" of the dates is to show the temporal range of the interferograms. We removed it; 3. The positive/negative error of the GL position is explained above; 4. The value is the distance between the downstream IGZ limit and the position used to calculate the tide. The location at 100 km is used. The reason of including the SSH at 50 km is to show the SSH does not significantly change from 100 km to 50 km; 5. The duplication has been removed. We have revised the figure to improve clarity.

8. Code availability

- The Zenodo link points to code used to compute basal melt rates at Thwaites, originally from a different paper (Gadi, 2024). I am not sure of the specific journal guidelines on this, but I suggest this should either be appropriately cited alongside the dataset, or specific code used for this work at Berry should be provided.

- No code is provided for the other calculations described in the methods, including ice bottom/bed topography, ice mass balance, GZ length or tides. This should be included to ensure full transparency.

Response: 1. Ratnakar Gadi is a co-author of this manuscript. The code is identical; 2. The calculations are done using the built-in tools and the Python console in QGIS.

Reviewer #2

Overall I think the manuscript contains exciting, new results, but the line of argument around the connection between the GL retreat of Berry and changes in CDW temperatures needs revision and refinement to make it convincing.

Generally, the manuscript needs some polishing. I suspect that shortening the text to make it fit within the Nature Communications format is the culprit here. This applies especially to the introduction that needs to be reworked – see my specific comments.

There are two main points in the manuscript. One is that you can identify tidal variability in the movement of the glacier and that this likely gives the warm CDW on the continental shelf access to the grounding line zone leading to high melt rates. The second one is that the grounding line of Berry has retreated rapidly over the past 25 years due to that CDW access.

My main problem here is that your line of argument focusses on the presence of CDW and not how it may have changed. There isn't much CTD data available, but what there is could be used better:

Show the temperature profiles that you are using and don't just leave the reader to eye-ball them from Jacobs et al. 2013. Also give data sources for the CTD data, especially the mysterious 2018 CTD that you base your heat flux calculation on. Use these temperatures and the position of the thermocline vs the associated GL positions and cavity shapes to discuss the implications for basal melt.

CDW access to the deep basin in front of Berry is most likely across the shelf break and not from the east on the shelf. The sill depth is shallower than for Siple Trough (see Jacobs et al. Oceanography 2012) and the off-shelf CDW core actually sits below the sill depth. This implies that it is likely even more episodic than we have shown for Siple Trough and even more dependent on winds along the shelf break and Ekman pumping (Assmann et al. 2019).

The implication of this is that you have a mechanism that explains why CDW access may have changed between 1994 and 2021: If you compare your mass balance time series in Fig. 4a with our Ekman pumping time series in Assmann et al. 2019 Fig 4f they agree rather well. You could calculate a similar time series for the shelf break in front of Berry, but it is unlikely that it would be much different because of the scales of atmospheric variability. Just like we argued that the enhanced Ekman pumping over the Siple Trough shelf break led to warmer CDW in front of the western Getz in 2016 & 2018 compared to 2000 and 2007 that argument could be used in this study as well.

Response: We thank the reviewer for the thoughtful and detailed feedback. We appreciate the recognition of the novelty of our results and agree that the argument linking GL retreat of Berry Glacier to changes in CDW temperature and access requires further refinement. We have made the following revisions in response to these suggestions:

1. We now explicitly show the CTD temperature profiles used in our analysis, rather than referring readers to previous figures (e.g., Jacobs et al., 2013). These profiles are presented in Supplementary Fig. 4b and 6b, with labeled data sources including a 2018 profile, which is from Korean Polar Center.

2. We revised the text to emphasize that CDW likely enters the deep basin in front of Berry Glacier from the continental shelf break, rather than from the east (Lines 181-183). We note that the CDW core offshore sits below the sill depth—implying episodic access, likely regulated by wind-driven Ekman pumping, as the reviewer rightfully pointed out (Lines 186-189).

3. We have incorporated the reviewer's suggestion to better connect CDW variability with changes in Ekman pumping. Specifically, we have added additional CTD profiles from 2014 and 2018 in the western Getz region, which exhibit clear signals of ocean warming (Lines 209-214, Supplementary Fig. 6b). These observations support the interpretation that enhanced intrusions of CDW—potentially driven by intensified Ekman pumping during periods such as 2016–2018—may have contributed to increased basal melt and the observed IGZ retreat. Thank you very much.

Specific comments:

- 1) Line 42: “affected” not “effected”.

Response: Fixed at Line 43.

- 2) Line 45: You are implying that changes in GL position affect sea level, because of the addition of seawater by floating ice. Floating ice shelves are hydrostatically balanced, i.e., they don’t affect sea level. What does affect is the mass flux across the grounding line and the mass loss from the ice shelves by basal melting and calving that adds freshwater, i.e., volume to the ocean. Just remove this sentence.

Response: Removed.

- 3) Line 50: This sentence is very out of context, move to last paragraph in the introduction where you introduce the location.

Response: Moved to Lines 70-72.

- 4) Line 52-53: “due to ocean tides” not “changes in oceanic tides”. Tides imply variability.

Response: Changed to “due to ocean tides” in Line 51.

- 5) Line 56-67: For my taste this is too technical for an introduction. I got lost in the detail and wasn’t sure why you are using all of these satellite data sources.

Response: The reason we use all satellite data sources is because ERS and ALOS-2 PALSAR allow us to look back in time, while Sentinel-1 and CSK provide high frequency repeat data to show the short-term GL dynamics. We noted this at Lines 57-63.

- 6) Line 66: You haven’t given any geographical context, i.e., the reader has no idea where or what Berry is. I assume this is result of shortening the text to fit within the Nature comms format.

Response: A short geographical context has been added at Line 64.

- 7) Line 71: Change “undiluted warm, salty CDW” to “warm CDW”. You don’t need salinity to make your point and I would remove it to simplify the story. CDW access on the western side of Getz is intermittent, so the CDW that makes it to the ice shelf front is anywhere from part of the thermocline to part of the CDW core (see Assmann et al. 2019).

Response: The sentence has been simplified in Lines 70-72.

- 8) Line 147-152: Duplicates the methods section, so you could remove this.

Response: Statement removed.

- 9) Line 161-162: Remove the reference to salinity. This is superfluous to your main storyline. Either write “mean bottom temperature of...” or change to something like “bottom temperature exceeding 1 C” to match the CTD sections in Jacobs et al. 2013 and sections and mooring records in Assmann et al. (2019).

Response: The sentence has been modified based on the comments at Lines 178-179.

- 10) Line 162-165: see comment about Supp. Fig. 4a and general comment about CDW access from the shelf break.

Response: Supplementary Fig. 4a is updated based on the comments to show the ice shelf front and the continental shelf break.

- 11) Line 165-167: This is the only mention you have in the entire article about changes in ocean temperatures and it is too hand-waving to support your main point. Show the actual temperature profiles in a figure, because it is hard to pin-point Berry in the Jacobs et al. (2013) figures.

Response: The numbers have been corrected in Line 185. The temperature profiles have been added to Supplementary Fig. 4b.

- 12) Line 181-185: I am not sure why you use 0.6 m as a thickness when you draw in 1.2 m worth of water volume over a tidal cycle.

Response: We assume the water thickness linearly decreases from +/-0.6 m at downstream (state 1) to 0 m at upstream (state 3), like a right triangle, therefore the water thickness of the entire IGZ is 0.6 m.

- 13) Line 186: The tide is diurnal at Berry (and generally in the Amundsen Sea), i.e., half a tidal cycle is 12 hours not 6 hours – see your Fig. 1.

Response: The value has been corrected to 12 hours at Line 207 in the revised text.

- 14) Line 187: Where is your 2018 CTD from? Add a source for all CTD data you use to your data statement. This is especially important for this 2018 CTD since you use the temperature for your heat flux calculations.

Response: We originally cited the number from Selley et al. 2021. But upon careful examination of the 2018 CTD dataset from the Korean Polar Center—used in the 2021 study—we found that the measurements extend only as far west as Dean Island and do not reach the vicinity of Berry Glacier. Consequently, to estimate post-2007 ocean temperature changes at the ice front of Berry Glacier, we analyzed CTD profiles collected from other locations in the west Getz Ice Shelf region with data available after 2007 (Supplementary Fig. 6). These CTD records indicate a general warming trend in the west Getz since 2000, which persisted beyond 2007, as evidenced by observations near Dean Island and east of Siple Island, which is consistent with your argument above that enhanced Ekman pumping over the Siple Trough shelf break contributed to the presence of warmer CDW in front of the western Getz Ice Shelf in 2016 and 2018 compared to 2000 and 2007. If the ocean in front of Berry Glacier experienced a comparable degree of warming, a thermal forcing of +3.6°C at 1,000 m depth represents a reasonable estimate. These data are now included in the discussion.

- 15) Line 190: I like the Gadi et al. (2023) reference a lot and agree that not all heat in the inflowing seawater is used to melt ice, but I interpreted the 30% statement in their study that 30% of the ice flux across the grounding line was melted in the GZ and FZ.

Response: Your interpretation is correct—the 30% figure refers to the fraction of ice flux melted in the grounding and flexure zones in the 2023 Gadi et al. paper.

- 16) Fig 1: Even with the improved resolution, panels in Fig 1 are still very small, this applies particularly to the top left insert in panel a.

Response: Figure 1 has been updated based on the comment.

- 17) Supp. Fig. 4 a: Add the ice shelf front to panel and extend the figure to the continental shelf break to show the channels giving the CDW access. In my experience, the bathymetry on the continental shelf around the western Getz isn't very detailed or reliable, because I don't think there is more multibeam data than shown in Nitsche et al. (2007). I would qualify the statement you make about these channels in lines 162-165.

Response: Supplementary Fig. 4 has been updated based on the comments. Cochran et al. use a geologically consistent inversion of NASA OIB airborne gravity data for bathymetry beneath the GIS, which extends for ~650 km along the Amundsen Sea coast of Antarctica, showing the existence of the thalweg.

- 18) Supp. Fig. 4 b & c: I do like the cavity sections, because they show nicely how the Berry GZ has moved to a retrograde slope, while the DeVicq GZ sits just at a small rise. I also suspect that bed shape is more responsible for the different reactions of the two glaciers rather than CDW temperature.

The temperature contours in panels b and c imply a thermocline at constant depth in front of Berry. I would have preferred a profile insert that shows that this is a point measurement. Showing would also give you the opportunity to show profiles for different years and to make your point about the CDW changes.

Response: Supplementary Fig. 4 has been updated based on the comments. We add the discussions related to the CDW and CTD from different years to Lines 196-199.

19) Supp. Fig. 5: I do wonder if it would be worth to move some of the panels into Fig. 5, because they illustrate the exceptional reaction of Berry within the region and give context to your discussion of DeVicq glacier which feels somewhat stand-alone in the manuscript.

Response: Supplementary Fig. 5 a&c has been added to Figure 5.

Reviewer #3

In this study, Chen et al. used DInSAR interferograms generated from different satellite data to map both the short-term and long-term grounding line migrations in Berry Glacier located in the Getz Ice Shelf from 1996 to 2021. They discovered a short-term grounding line migration over 18 km and claimed that the glacier bed is up to 1300 m deeper than the BedMachine data. The methods used in this study are standard and not novel, particularly for research focused on grounding line mapping using DInSAR. The identification of a large 'Ice Grounding Zone' in Berry Glacier is also not new, as it was previously discovered by Mohajerani et al (2021). The paper does not offer significant advances in our understanding of either short-term grounding line migration or long-term changes related to ice dynamics, as these concepts have already been extensively explored and established in previous studies. Furthermore, the interpretation of bed topography and the calculation of short-term grounding line migrations require further consideration and improvements.

Major Comments:

- 1) The finding of the large tidally-influenced grounding line migration in Berry Glacier is not new, Mohajerani et al (2021) have already found grounding line changes in Berry Glacier to be about 15 km (<https://www.nature.com/articles/s41598-021-84309-3>). Given this, I feel this paper only provides a more detailed calculation building upon the work of Mohajerani et al (2021), and its main conclusion lacks novelty. Especially, Mohajerani et al (2021) have analyzed the entire Getz Ice Shelf. I am curious why the authors in this study chose to focus solely on a single glacier within the same region. Including all glaciers in Getz Ice Shelf would allow for a comparison of grounding line changes across glaciers that share similar oceanography settings, this would result in a more comprehensive study. Focusing on just one glacier significantly limits the scope of this research.

I agree that short-term grounding line migrations have not been adequately incorporated into ice sheet modeling. However, this phenomenon has been well-documented in Antarctica in recent years using a similar DInSAR approach and datasets to those used in this study, with some examples listed below. While I acknowledge the efforts presented in this paper, I feel that its conclusions do not significantly advance our understanding of short-term grounding line migrations. Instead, the study only adds incremental evidence to the existing body of literature.

- Milillo 2017:
<https://agupubs.onlinelibrary.wiley.com/doi/full/10.1002/2017GL074320>
- Milillo 2019: <https://www.science.org/doi/10.1126/sciadv.aau3433>
- Brancato 2020:
<https://agupubs.onlinelibrary.wiley.com/doi/10.1029/2019GL086291>
- Mohajerani 2021: <https://www.nature.com/articles/s41598-021-84309-3>
- Milillo 2022: <https://www.nature.com/articles/s41561-021-00877-z>
- Chen 2024:
<https://agupubs.onlinelibrary.wiley.com/doi/10.1029/2022GL102430>

Response: We appreciate the reviewer's thoughtful comment and acknowledge that Mohajerani et al. (2021) was the first to document the presence of a large tidally-influenced IGZ in Berry Glacier using Sentinel-1 DInSAR. Our study builds upon their important finding by providing higher-quality interferometric observations and a more detailed spatiotemporal analysis of the evolution and mechanisms behind the observed GL behavior.

Specifically, the shorter wavelength and repeat cycle of Sentinel-1 limits coherence in interferograms over fast-flowing ice near GLs. To overcome this, we incorporated CSK data, which offers high temporal resolution and superior coherence in the IGZ. This enabled us to more precisely resolve GL migration patterns, tidal flexure, and transitions between different grounding states. CSK coverage is limited spatially, and unfortunately, we do not have comprehensive CSK data across the entire Getz Ice Shelf.

We chose to focus on Berry Glacier because it exhibits unique dynamic behavior not seen in other Getz tributaries. Unlike nearby glaciers, Berry hosts a deep, wide IGZ with distinct tidal signatures, and a complex bed geometry that allows—but also limits—warm CDW penetration. Through analysis of elevation time series, interferometric phase patterns, and reconstructed basal topography, we identify the processes by which CDW accesses the IGZ via a thalweg and how melt efficiency is modulated by subglacial bed conditions. These findings, including the role of trapped seawater and temporally evolving tidal states, are not presented in previous studies.

Therefore, while our study builds on earlier DInSAR-based GL mapping, it contributes new insight into the ocean-ice-bed interaction at fine spatial and temporal scales, and highlights processes critical to improving ice sheet model representation of short-term grounding line variability.

- 2) The comparison between 1996 IGZ and 2020-2021 IGZ needs more care. In 1996 you only have two interferograms, it is very likely due to limited DInSAR data acquisition during that time, this may have resulted in incomplete capture of short-term grounding zone migrations. While in 2020-2021, there is a greater availability of data, enabling greater sampling and a more comprehensive imaging of the entire IGZ. Please address this in the analysis.

Response: Yes, in the revision we note the following. In the two 1996 interferograms, we see no residual signal upstream of the IGZ. In more recent interferograms, there is always a residual upstream, indicative that during some part of the tidal cycle, seawater intrudes farther up. It is therefore unlikely that seawater intruded much farther in 1996. Secondly, as shown in the study, the glacier thinned considerably since 1996 (11 ± 1 m/yr). In particular, ice deviated from flotation by 1 of meters at the location of state 2, which the grounding line reached around 2006. See Lines 277-280. The IGZ in 1996 could not have extended to state 2.

- 3) Figure 2a shows the spatial density of different grounding line positions obtained in this study, capturing three different states. However, a limitation of this approach is the lack of consideration for temporal sampling. If the GL acquisitions are concentrated around same tidal phases, the resulting spatial distribution map may be biased. In addition, the gap between State 2 and 3 is too short (~ 1 km), making this classification unreliable.

Response: We appreciate the reviewer's comment and agree that temporal sampling and tidal phase distribution are important considerations when interpreting GL positions. To address this, we note that Fig. 2d demonstrates the presence of all three GL states under both positive and negative tidal phases, suggesting that the classification is not an artifact of biased tidal sampling.

Regarding the distance between state 2 and state 3, the spatial separation is approximately 2 km, rather than ~ 1 km. While the gap is relatively narrow, we argue that the two states are physically distinct. Specifically, the presence of reversed motion between states 2 and 3 implies the existence of subglacial space available to accommodate trapped seawater during tidal cycles. This behavior supports our interpretation that states 2 and 3 represent different stages in the evolution of grounding line migration, rather than minor fluctuations within a single transitional state.

We have also considered the influence of spring–neap tidal cycles and the distinction between ebb and flood tides (Supplementary Fig. 1); however, no discernible correlation between tidal phase and GL migration is detected. This is now included in the paper.

- 4) It is interesting that the authors found the bed topography underneath Berry Glacier to be 1300 m deeper than the BedMachine. This is useful information that can be used to update the BedMachine dataset. However, I am not particularly surprised that the Bedmachine dataset is inaccurate in some of the glaciers due to a lack of observations.

Additionally, it is unclear how this discovery of deeper bed topography changes our current understanding of the grounding line dynamics and regional mass imbalance.

Response: There are numerous examples of poor delineation of bed depth in the OIB data because the radar echoes from the bed are sometimes difficult to pick. This study will indeed have an impact on BedMachine in this region.

- 5) The discussion about seawater intrusion is unclear. In Line 139: the authors claim that ‘A series of bed depressions between the three states must trap seawater...’ I feel this claim is not well supported and more evidence is needed to make sure this statement is scientifically correct. Although the authors claimed to detect the reverse fringes of opposite motion upstream to indicate the trapped seawater, there are not enough details on how the water extent is mapped in Figure 3 and how the presence of trapped seawater changes in sync with grounding line migrations.

Response: The extents of the three GL states were determined based on DInSAR-derived GL positions. We infer the presence of seawater in the gaps between these states—specifically between state 1 and state 2, and between state 2 and state 3—based on two lines of evidence: 1. Fringe patterns of reversed motion in Figure 2b: The locations of reversed interferometric fringes, which indicate vertical deformation, spatially coincide with the regions between GL states. This consistency supports the interpretation that these zones are periodically flooded with seawater during high tides and partially drained during low tides, consistent with the flexural response of floating ice over subglacial water; 2. Elevation difference between ice draft and radar-derived bed: Along profile C–D, the 2012 MCoRDS-derived bed is significantly deeper than the 2019/2020 flotation-derived ice draft, indicating the presence of sub-ice water. However, we acknowledge that this evidence is limited to specific locations, and that we cannot constrain the full spatial extent of the deeper bed or the thickness of the trapped water in other regions.

- 6) Mismatching patterns in ice dynamics: The dhdt in the west flank is much lower than the main glacier trunk where there is an over-deepened retrograde bed slope, this does not match the basal melt rate patterns in Figure 5 and the ice velocity changes in Figure 3. Based on MISI theory, we would expect to see matching patterns of ice thinning, grounding line retreat, as well as ice speed up along the overdeepened bed. The authors, however, only provide one comment on this mismatching pattern in Line 114–115: ‘The pattern of thinning does not coincide with the pattern of speed up as the retreat expands laterally along a tributary’. This explanation is too simplistic and misleading. Please provide a more detailed explanation.

Response: The apparent mismatch noted in the manuscript refers to the western region outside the IGZ, where we observe substantial surface lowering without a marked increase in ice velocity. Within the IGZ, we have consistent patterns of GL retreat, ice thinning, and speedup, in line with the MISI theory. On the west of the IGZ, the discrepancy may be due to a combination of factors, including: 1) limited driving stress due to thinner ice; 2) lower original speed (tributary), any speed up will be more difficult to detect; or the observed thinning is influenced by SMB processes rather than dynamic thinning. However, in the revision, we removed the comment about the coherence between speed up and thinning.

Specific Comments:

- 1) Line 51–55: the jump from basal resistance associated with the grounding line to the lack of consideration of short-term grounding line migration in glacier modeling feels out of place here. Please consider talking about the importance of short-term grounding line and its observations across Antarctica first.

Response: Our intent was to emphasize that short-term GL migration plays a critical role in modulating the basal friction regime, which directly impacts glacier velocity, ice discharge, and ultimately mass loss—key parameters in numerical ice sheet models. Therefore, recognizing and incorporating short-term GL variability is essential for

improving model fidelity.

- 2) Line 57-59: the accuracy of GL mapping in altimetry and DInSAR needs references

Response: We have added appropriate references in Lines 56-57 to support the stated accuracy of grounding line mapping: Zwally et al. (2002) for altimetry-based detection and Rosen et al. (2000) for DInSAR-based methods.

- 3) Line 97: 'GL is predominantly in state 1 in 1996...' also depends on the number of DInSAR interferograms and the tidal samplings.

Response: As we mentioned above, in the two 1996 interferograms, we see no residual signal upstream of the IGZ. In more recent interferograms, there is a residual upstream, indicative that during some part of the tidal cycle, seawater intrudes farther up. Therefore, we may underestimate the length of 1996 IGZ due to the limited number of interferograms, but it was unlikely that seawater intruded farther than state 1 in 1996.

- 4) Line 129-130: 'up to 1300 m lower than those reconstructed in BMv3.7', could you explain more how this number is obtained in the figure or in the main text?

Response: We computed the difference between the flotation-based ice draft within the IGZ and the bed elevation from BM v3.7 to quantify the discrepancy. The maximum difference of 1300 m occurs on the eastern side of the IGZ, in a region that does not coincide with any radar profiles (Lines 140-143).

- 5) Figure 1: please increase the size of the inset map of Getz ice shelf in subplot 1a, it is difficult to see the details in its current format.

Response: Figure 1 has been updated based on the comments.

- 6) Figure 3: please avoid using rainbow colormap in line plots 3c/d/e/f, it is hard to distinguish between different years, perhaps a sequential colormap is better. In 3g and h, please clarify the question mark annotated on the figure in the caption, this part is also unclear from the main text.

Response: 1. Our initial choice was intended to maintain consistency with the color scheme used for grounding line (GL) positions. To improve clarity, we have modified the colorbar to better highlight the annual trends in surface elevation and velocity. Additionally, we now provide an alternative version of the figure using a different colorbar; 2. The information regarding the shaded areas and the meaning of the question mark annotation in panels 3g and 3h has been added in the figure caption and the main text (Lines 359-364).

7) Line 151: why 4.5 km?

Response: We test different window sizes for the filtering until the melt rates do not change at a detectable level and heterogeneities are eliminated. A window size of 4.5 km is consistent with the expected width of the bending zone, i.e. the zone where the glacier is pushed below and above HE, as per Chartrand and Howat, 2024.

8) Line 160: could you map or annotate the boundaries of this surface elevation bumps in the satellite images in Supplementary Video 1? It is difficult to distinguish this feature in the current video.

Response: Agreed. We have added circular annotations in Supplementary Video 1 to highlight the locations of the surface elevation bumps. The circles are intentionally drawn larger than the bumps themselves, as precisely mapping the boundaries would also include surrounding surface undulations, which obscure rather than clarify the features of interest.

Dear Editor, Reviewers,

We greatly appreciate the reviewers' thoughtful and constructive comments again, which have helped strengthen the manuscript. We hope that the revised version will meet the expectations of the reviewers and that the paper will be found suitable for publication.

On behalf of the co-authors – Hanning Chen.

Reviewer #1

General Comments:

I commend the authors for their thorough revision of this manuscript, and I agree that it is now significantly improved! The title is much improved and the majority of my concerns have been addressed, regarding the text, methods descriptions, and the treatment of uncertainties. The figures are in much better shape too. I have one major remaining concern regarding the discussion about the cavity formation (detailed below), and a few additional minor comments / suggestions that the authors may like to consider before final submission. Once addressed, I would recommend this manuscript is acceptable for publication.

I commend the authors for their thorough revision of this manuscript – it is now substantially improved. The revised title is clearer, and the authors have addressed the majority of my previous concerns regarding the text, methodological descriptions, and treatment of uncertainties. The figures are also much improved. I have one remaining major concern related to the discussion of cavity formation (detailed below), along with a few additional minor comments and suggestions for the authors to consider prior to final submission. Provided these points are addressed, I would recommend the manuscript for publication. Congratulations.

Response: We greatly appreciate the reviewer's positive and encouraging feedback on the manuscript. We have carefully addressed the remaining concerns as outlined in our detailed response. Thank you again for your constructive guidance throughout the process, which helped us improve the clarity and robustness of the study.

Major Comments:

Discussion of the Berry Cavity Formation

I appreciate the additional explanation made in the response document regarding the interpreted formation of a >100m deep cavity in the Berry IGZ, however I don't think this has been sufficiently reflected in the revised manuscript. In your response, you state:

“the cavity has areas of partial grounding, and the water column must be thicker in between these pinning points, perhaps 10's m, but we do not have data to quantify it”.

However, this doesn't seem to match up with what is written in the manuscript; i.e.,

L172 ‘During the past 25 years, a cavity several 100 meters in size has formed beneath Berry, up to 300 m near state 2’;

L205 ‘From the modeled tidal height of ± 1.2 m (Fig. 2d), we estimate that the average water in the IGZ cavity must be at least 0.6 m’

L226 “Most likely, the thickness of the water column coming in and out of the cavity must be greater than 0.6 m, possibly by a factor of 2.5”.

Perhaps this discrepancy is related to your comment made in the response document:

“The DInSAR fringe patterns observed in this region reflect the vertical displacement of the ice surface driven by subglacial water motion rather than the total water thickness”.

I would argue that this distinction is still not made clear enough in the manuscript: Is the 0.6 m value relating to the change in depth in the cavity as measured by the DInSAR, on top of an absolute cavity depth of >100m? And how does this relate to the pinning points? To address this before publication, the authors should:

1. Reconcile these estimates for cavity depth, both the distinction between the 0.6m and >100m values, as well as whether the cavity depth should be on the order of 10s or 100s of meters.
2. State that you don't have the data to accurately quantify the depth of the cavity (as you say in your response).
3. Soften the language used to describe your interpretations of the cavity formation (i.e. make it clear that the observations suggest that a cavity >Xm thick has formed at state 2, rather than stating it as definitive fact.).

Response: We only have direct measurements of bed elevation—and thus reliable estimates of cavity depth—along a few OIB flight tracks. Outside of those tracks, the DInSAR-derived vertical ice displacement measures the tidal motion of partially grounded ice, not to the absolute height of the water column beneath the ice.

We estimate the minimum volume of seawater that enters the cavity during tidal cycles by assuming a change in cavity height that tapers linearly from a maximum height of 1.2 m at the entrance (maximum sea surface height minus minimum sea surface height) to zero at the inland limit of the grounding zone, hence an average height of 0.6 m. The true cavity height is likely significantly greater. Indeed, we find that the ICESat-2 derived surface thinning are higher than that expected from the ocean heat content of this intrusion, i.e., a much larger volume of seawater must access the cavity. We clarify this very important point at Lines 231-234 and Lines 186-190. We added one sentence to summarize our findings on the bed topography to make it more clear at Lines 159-163.

Minor Comments:

Use of HE for calculating thickness and basal melt

Thank you for explaining your application of hydrostatic equilibrium for ice thickness and basal melt estimates. In particular, the following response was very helpful for understanding and justifying your approach, and should therefore be included in the methods section of the paper. “We apply the hydrostatic equilibrium (HE) assumption only to regions where seawater intrusions are inferred at a specific time period. For the period 1996–2005, changes in ice thickness were equated directly with surface elevation changes, as the GL reached to state 2 around 2006. Between 2006 and 2011, the HE assumption was applied within the region from state 1 to state 2. From 2012 to 2022, HE was applied across the entire 2021 IGZ extent.”

Response: The description has now been added to the methods section at Lines 350-355.

Specific Comments:

L205-225 - I appreciate the edits to this section speculating on max melt rates - it is much stronger now. In your response to my initial comments, you state: “This calculation is a maximum melt rate if water intrusions flush the entire cavity and all available heat is used to melt the ice”. This is a critical statement that should be included in the text here - either at the very start of this paragraph or in the last few sentences.

Response: We now use a more realistic assumption that only $30 \pm 10\%$ of the available ocean heat is used to melt ice, see Line 240. We emphasize that this calculation is affected by our limited knowledge of the cavity thickness, see Lines 243-247. We also noted that this calculation indicates that the true thickness of the water intrusions may be an order of magnitude larger at Line 247.

L226-240 (broader implications) - This new text is a valuable addition to the paper. I have a couple of minor suggestions for clarity:

- I suggest rewording the first sentence to something like: ‘While the relatively small ice volume of Berry Glacier limits its potential sea level rise contribution, our results have important implications for ice sheet modeling in Antarctica’

Response: Done at Lines 248-250. Also added its importance to the mass loss of the Getz Ice Shelf area at Lines 250-251 and 141-144.

- L233 - I am not clear on what ‘thinning the strand’ means. Please clarify.

Response: We re-wrote that the increase in melt forced the GL to retreat, reducing the basal resistance to flow, allowing the glacier to accelerate, which is conducive to thinning and further retreat. See Lines 256-258.

L376 - I suggest you should add the additional information about the 4.5 km spatial filter that you provided in your response to the manuscript text at L376-377 (‘The window size is 4.5 km, which roughly corresponds to the size of the bending zone discussed by Chartran et al. 2024’)

Response: We added a reference to Chartrand and Howat at Lines 404-406 in the methods section.

Figure Comments:

Fig. 3

- Panels G and H are much clearer and easier to interpret now! A couple of minor remaining points:
 - Still missing legend showing what stripey blue areas is
 - I presumed that the shaded blue area was error bounds on the IS2 and WV-derived ice bottom elevations, but perhaps I am mistaken. In the methods (L359) you write “The blue shaded areas in Fig. 3g mark the regions where vertical motion from seawater intrusion is detected, but the bed depth is unknown” – this is a very helpful description that is important for the interpretation of the figure, so I suggest you include a similar sentence in the figure caption to clarify this for readers.

Response: 1. A legend for the stripey areas has been added; 2. Agreed. The "blue shaded areas" mentioned in Line 359 refer to the blue stripey regions in Fig. 3g, which indicate locations where vertical motion from seawater intrusion is detected, but the bed depth is unknown. This has been clarified and corrected at Lines 386-388. The blue shaded areas in panels (g) and (h) represent uncertainty bounds on the ice bottom elevations derived from ICESat-2 and WorldView data. The relevant text has been added to the figure caption.

Fig. 5

The addition of white boxes showing the “area of coincident measurements” is very helpful here! Please make sure this is sufficiently described in the caption though.

Response: Agreed. The relevant descriptions have been added in the caption of Fig. 5.

Supp. Fig. 7

- The additional information about tides on each panel is very useful.
- I suggest labeling each panel with the date e.g. 1996a, 1996b, 2008a etc, so this ties in more directly with the table below.

Response: The date labels have been added to each panel of Supplementary Fig. 1 (used to be Supplementary Fig. 7).

Reviewer #2

In general, the whole manuscript reads more balanced and is greatly improved after the authors

incorporated mine and my fellow reviewer's comments.

The discussion of the influence of the CDW and its recent changes is now much better. Especially Supp Figs. 4 and 6 reflect the hydrography and data base much better.

As a minor revision before final acceptance of the manuscript I would like to see some re-organisation of the discussion of the influence of CDW in the main manuscript.

Response: We thank the reviewer for the positive and constructive feedback. In response to the suggestion for improving the structure of the CDW discussion, we have reorganized the relevant sections in the main text to follow a more logical progression, as guided by the reviewer's specific comments. Specifically, we now first introduce the direct CTD observations at the ice front of Berry Glacier, followed by a discussion of recent temperature trends based on CTD data from 2018, and link these changes to enhanced Ekman pumping and the inferred CDW pathways. See Lines 195-214. We believe that this revision will improve the clarity and impact of the discussion as suggested by the reviewer per below.

General Comments:

Line 178-189 and 208-214 which contain the main discussion on CDW are somewhat repetitive and would benefit from merging and re-ordering. I suggest doing this following your very nice figures Supp. Fig. 4 and Supp. Fig. 6.

Supp. Fig. 4 shows that there are direct observations of CDW at the ice shelf front of both Berry and duVicq with some warming between 1994 and 2007. Use these to discuss access of the CDW to the GL and the importance cavity shape.

Then use Supp. Fig. 6 to take in the more recent observations and the CDW warming to present day. Since these are not located directly at Berry, this is the place to discuss CDW pathways on the continental shelf and the enhanced Ekman pumping at the shelf break as the mechanism that controls CDW access to the continental shelf and thus the warming. It is worth mentioning that the CDW at Berry may not have reached 1.2 C in 2018, since CDW there appears to have been cooler than at Dean in the earlier years.

Response: We have rewritten the text per these recommendations starting at Lines 195-214. We note at Lines 235-236 that the "1.2°C" is inferred from 2018 CTD observations near Dean Island due to a lack of data at the front of Berry.

Specific Comments:

l.181: remove "because they are deep enough" Also since this refers to the bathymetry on the continental shelf outside of the ice shelf cavities that figure reference should be Supp. Fig. 6.

Response: The sentence has been removed. The reference to the supplementary figure has been corrected at Line 200.

l. 208-209: "The ocean temperature on the western Getz Ice Shelf" should be "The ocean temperature along the western Getz Ice Shelf"

Response: The sentence has been changed and moved to Lines 203-204.

l. 236-238: This sentence is unclear. Re-write.

Response: The sentence "The evolution of Berry Glacier is illustrated by the feedback between newly formed cavities and thermal ocean forcing in melting grounded ice, whereby a complex cavity shape can reduce the efficiency of ice melt" has been rewritten "However, the complex shape of the newly formed cavity, which includes ridges, over-deepening, and areas of partial grounding, must have limited the amount of ocean heat that can flood the cavity, and in turn moderate the rate of glacier retreat" at Lines 258-260.

Supp Fig 6: Remove the CTD stations east of Siple. CDW there doesn't have direct access to

your region of interest and this would make the figure easier to read.

Response: The figure has been modified and the relevant description in the text has been removed.

Reviewer #3

Thank you for addressing my comments and revising the paper. I only have some minor suggestions below.

Line 71: Please consider rephrasing the sentence into ‘are in contact with warm waters coming from Circumpolar Deep Water that ...’

Response: The sentence has been rephrased “Near the GL of Getz, many glaciers are in contact with warm waters coming from Circumpolar Deep Water (CDW), which rapidly melt ice from below.” at Lines 70-72.

Line 72: The mention of ice shelf meltwater feels out of place here. I suggest adding a statement on the implications of ice shelf basal melting.

Response: Agreed. The sentence has been rewritten “As a result, the Getz Ice Shelf is one of the largest producers of basal meltwater in Antarctica, which contributes to weakening ice shelf buttressing and enhancing dynamic mass loss.” at Lines 72-74.

Figure 1a: It’s difficult to discern the black arrow associated with Berry Glacier from the background ice velocity map. I suggest changing to a different color. The same issue applies to the star marker at the tide location. Also, the star marker is very small.

Response: Agreed. The figure has been improved based on your comments.

Line 78: ‘but there is no residual signal upstream of state 1 in 1996’, please add a reference to Figure 7.

Response: The reference to Supplementary Fig. 1 (used to be Supplementary Fig. 7) has been added at Line 80.

Line 83-86: Please specify which bed topography product you are referring to.

Response: We refer to BedMachine v3.7. This has been clarified at Line 87.

Line 172: Please rephrase ‘several 100 meters’

Response: The sentence has been rephrased at Lines 186-189 to emphasize that we have no data to constrain the cavity shape away from the OIB flight lines. We note however that the thinning rates imply a water column of several hundred meters must have formed over time between state 1 and state 2, see Lines 162-163.

Line 194 - 201: I can’t tell changes in basal melt rates in different time periods or different locations mentioned in this paragraph directly from Figure 5, as the current colorbar of basal melt rate has a limit of [-6,6] m/yr. Please revise the figure to improve the illustration of basal melt rate variations.

Response: The color bar range has been updated to show the variations better in Fig. 5.

Line 204: Consider removing the repetitive ‘that is’ in this sentence.

Response: Done at Line 229.

Dear Editor, Reviewers,

We are grateful for the reviewers' thoughtful, constructive comments in the whole revision process, which have substantially improved the manuscript and brought it into compliance with publication standards.

On behalf of the co-authors – Hanning Chen.

Reviewer #1

General Comments:

Thank you to the authors for their thoughtful responses to my comments and for the efforts made to address these in the revised manuscript. I am satisfied with their responses and with all changes made. The additions to both the text and figures have greatly strengthened the manuscript. Congratulations to the authors.

Response: The reviewer's constructive guidance at every stage significantly improved the study's clarity and robustness, for which we are sincerely thankful.

Minor Comments:

I have two very minor residual comments that the authors may wish to consider before publication:

- L140 - Missing unit on " 13 ± 15 " for the SMB.

Response: The unit is added.

- Figure 3 – Thank you for clarifying these points in your response and the revised manuscript. To avoid any potential confusion, I suggest using a different color for the blue striped regions (locations where vertical motion from seawater intrusion is detected) and the blue shaded regions (ice-bottom uncertainty bounds). At present, the two blues are difficult to distinguish, particularly where they overlap.

Response: We changed to two distinguished blue colors to avoid the confusion.

Reviewer #2

I am satisfied with the author's response to my comments and the resulting edits in the manuscript. I hereby recommend publication of the manuscript.

Response: We are deeply grateful for the reviewer's constructive guidance throughout the process, which sharpened the clarity and strengthened the rigor of our study.

Review of Chen et al. (2024) Rapid retreat of Berry Glacier, West Antarctica, from satellite radar interferometry and other data.

Summary: This study uses a wide range of satellite observations from 1996-2022 to document changes in grounding line (GL) position, tidal grounding zone (GZ) dynamics, ice velocity, discharge and mass balance of Berry Glacier, which feeds 10% of the Getz Ice Shelf in West Antarctica. The authors report substantial increase in the length of Berry's ice grounding zone (IGZ), from 0.6 km in 1995/96 to 18 ± 0.9 km in 2020-21. Over the same period, they report changes in ice dynamics, including an estimated 17 ± 1 m/yr of thinning, a $64 \pm 5\%$ increase in ice flow speed, and a total mass loss of 131 ± 23 Gt. The authors present novel observations of short-term GZ dynamics within the Berry IGZ show that the GL migrates between three states, independently of tidal forcing, suggesting complex interactions between tides and GL movement that are not adequately captured in current ice sheet models. These short-term dynamics are proposed to facilitate seawater intrusions, driving the increased basal melt, thinning, and ice discharge. Furthermore, the authors propose that the bed topography within the Berry IGZ is up to 1,300 m deeper than indicated by MCoRDS airborne radar.

General Comments: This work has impressive scope, integrating a wide range of satellite datasets over Berry Glacier – a region that has not previously been the subject of such a comprehensive study. I believe the results of the study will be of interest to the community, particularly regarding the novel observations of tidal seawater intrusions and their potential impact on basal melt rates within grounding zones. However, I have several significant concerns that should be addressed by the authors before publication. These concern the lack of robust uncertainty estimation or discussion of limitations, as well as certain aspects of the methodology and data interpretation that may affect the validity of the conclusions. It is crucial that the distinction between *observations* and *interpretations* is made clear throughout the manuscript. Additionally, the text and figures require refinement in many places to bring them up to publication standard. Below, I outline my major concerns below, followed by a list of specific suggestions for improving text clarity, figures, code and datasets.

Originality and Significance: The study provides new insights into the dynamics of the IGZ at Berry Glacier and the first published observations of changes in ice velocity, discharge and melt rates over the past 25 years. However, it should be noted that Berry Glacier is relatively small in the context of the Antarctic Ice Sheet. Thus, while the findings are significant for this setting and contribute to improved process-level understanding, their broader implications for ice sheet modeling and sea level rise projections are potentially more limited. I recommend enhancing the discussion of these broader implications, addressing—for example—how findings at Berry compare to those from other Antarctic IGZs and how they might contribute to modelling improvements on an ice-sheet-wide scale.

Treatment of uncertainties: There is an insufficient and inconsistent treatment of errors and uncertainties throughout the analysis, including in the main text, methods and attached dataset. I strongly recommend that the authors include a more comprehensive discussion of the uncertainties and errors in each part of the analysis and how these propagate through the

calculations. I recognize that quantifying some of these uncertainties may be challenging; nevertheless, an open discussion of these limitations is essential to avoid potentially misleading readers or overinterpreting the results.

Robustness of Conclusions: I have some concerns about the validity of some of the data interpretation that form the basis of the paper's main conclusions. I would like to see a response from the authors providing further clarification or justification of these.

1. *Long-term GL retreat 1995-2021*

The authors report that the length of the Berry IGZ increased from 0.6 km in 1995/96 to 18 ± 0.9 km in 2020-21. The 1995/96 IGZ length seems to be based on data from just two ERS1/2 interferograms, compared to >15 interferograms for 2020-21. I appreciate that there is limited data availability from 1995/96, but I wonder whether the reported shorter IGZ length in the 1990s may be influenced by the tidal sampling resolution. If the two ERS1/2 interferograms only captured the GL when it was in its Stage 1 position, it's possible that any inland migration of the GL during other tidal phases was missed. This could lead to an underestimation of the IGZ length during this period, and therefore an overestimation of the long-term change by 2020-21. I recommend that the authors discuss this potential bias in the interpretation and clarify how tidal sampling might impact the results (1-2 sentences would suffice). Providing the modelled differential tide and maximum sampled tide for each of the ERS1/2 SAR scenes in both Supplementary Figure 6 and in the data spreadsheet (as for the Sentinel-1 and COSMO-SkyMed data) would provide valuable context.

2. *Calculations of ice thickness using HE*

The authors compare the GL retreat at Berry to observed changes in velocity and ice elevation from 1996-2022 and estimate how this translates to changes in ice thickness, discharge and total mass balance. I am concerned about the validity of using hydrostatic equilibrium (HE) to convert ice elevation change observations to thickness changes. Additional justification of this approach is needed, along with a more open discussion of the associated limitations and uncertainties in the main text.

While using HE for this calculating ice thickness is appropriate for fully floating ice, the InSAR data indicate that the ice in the Berry IGZ is grounded for a significant portion of the tidal cycle, with an average depth of seawater intrusions of ~0.6 m. This implies the ice within the IGZ cannot be in hydrostatic equilibrium, casting doubt on the validity of ice thickness values derived from this method, and, by extension, the estimates of thinning rates, ice discharge, basal melt rates and bathymetry. For example, the authors' interpretation that a 24 m surface elevation decrease in the IGZ translates to 222 m of thinning (L101-102) is likely to be a significant overestimation.

It is well-documented that satellite-derived estimates of ice thickness and basal melt rate within GZs are subject to large uncertainties due to the breakdown of the hydrostatic assumption (e.g., Chuter & Bamber, 2015; Griggs & Bamber, 2011; Kim et al., 2024), which often leads to these

regions being masked out (e.g., Griggs & Bamber, 2011; Paolo et al., 2015; Wilson et al., 2017) or interpolated across (e.g., Fretwell et al., 2013; Morlighem et al., 2020). While I note that some attempt has been made by the authors to address this (e.g. L150-152; 296-298), the discussion remains too vague and does not adequately address the impact on the key findings.

3. *Refinement of bed topography*

The authors propose that the bed topography within the Berry IGZ is up to 1,300 m deeper than measurements from MCoRDS airborne radar and in BedMachine v3.7 (which interpolates between sparse radar measurements). However, I do not think that the evidence presented here is strong enough to support this conclusion; I suggest the authors either reconsider this interpretation or provide a more robust justification, clearly communicating any limitations. Specific concerns include:

- a) **Methods description:** The description of methods used to reconstruct bed topography lacks sufficient detail, both in the main text (L125-127) and methods (L278-279). I recommend expanding the description to clarify the approach, including details on the geoid and specifying the areas or flowlines over which bed topography was calculated.
- b) **Use of HE/flotation:** If the new bathymetry estimates in the IGZ are based on ice thickness calculated using HE/flotation, they are subject to the same uncertainties outlined above. Prior studies have demonstrated that ice thicknesses calculated in this way close to the GL tend to overestimate values compared to radar measurements due to the breakdown of HE (e.g. Chartrand & Howat, 2023).
- c) **Radargram interpretation:** The authors propose that the bed depths from the MCoRDS radar in the Berry IGZ have been misinterpreted as too shallow (L132-134). However, upon reviewing Supp. Figure 2 (noting that I am not a radar specialist), I find this interpretation unconvincing. While weak bed echoes at the glacier center may well indicate higher uncertainty in bed depths, the picked bed depths (red dashed lines) in all four radargrams look reasonable and there is little indication of any deeper echoes at the depths suggested by the authors. Radar is generally considered most reliable way to measure ice thickness/bed depth in these regions, and it seems more plausible that the HE-based calculations in this study might overestimate thickness/depth compared to the radar. Labelling specific features in the radargrams that support the deeper bed interpretation would be helpful, as well as providing a comparison to data from the gravity inversion mentioned in L136.
- d) **Clarification of the 1300m depth difference:** The origin of the 1,300 m value is unclear. In Figure 3(g-h), the difference between the BedMachine bed elevation and the author's new calculations appear to max out at around 1,000 m. Since this 1,300 m value is presented as a key result in the abstract, its basis should be clarified in the text.

4. *Long-term melt rates / basal melt regime*

The text in this section (L146-192) needs refining; it is currently difficult to follow the specific areas and time periods for which melt rates have been calculated. Clear distinctions are needed between discussions of: (a) Melt rates within the Berry IGZ (2019-2022); (b) Melt rates within a

zone 10 km seaward of the IGZ, compared to neighboring glaciers and previous measurements; and (c) Predicted melt rates within the Berry IGZ due to tidal seawater intrusions

Again, my concerns about the use of HE to calculate thinning rates within the IGZ alongside the lack of uncertainty quantification make me skeptical about the validity of the derived melt rates. The authors briefly mention using 4.5 km spatial smoothing to mitigate deviations from hydrostatic equilibrium (L150-152), but further detail is needed to explain what was done and how this reduces uncertainty. As I understand, the authors have interpreted the formation of a >100m thick cavity beneath the Berry IGZ over the past 25 years (L156), reaching depths up to 300 m near State 2. This implies the whole region is afloat, which starkly contradicts the InSAR GL results which show the whole IGZ is ephemerally grounded with depths of seawater intrusions averaging just 0.6 m. Clarification of this discrepancy is important.

5. *Tidal seawater intrusions*

The estimates of melt rates due to tidal seawater intrusions (L181-192) appear to rely heavily on assumptions and values that are either poorly constrained or lack clarity on their uncertainties. This undermines confidence in the results and may mislead readers, especially if the values are subsequently used for in models. It is recommended that the authors clearly outline all assumptions, cite where the values are taken from (note specific comments below about the CTD data), and acknowledge the scale of the limitations.

Diversity, Equity and Inclusion: I would like to draw attention to the lack of female representation among the nine co-authors, which might be something the authors or the journal could consider addressing in future to foster more diversity in scientific collaborations.

Specific Comments

Title – The title doesn't really reflect the key findings of the paper; I suggest this could be updated to include something on seawater intrusions which I think are arguably the most interesting result. Also 'and other data' is a bit weak.

Abstract – should be updated to reflect any changes made to the manuscript during revision.

L42 – typo; 'affected'

L47-49 – These sentences are quite vague. Are you referring to the effect of short-term GL retreat and advance periods (i.e. with the tides), or longer-term retreat/advance phases. Please clarify and consider adding references for support.

L51 – This paper doesn't directly discuss CDW; you might consider citing more relevant oceanography papers, such as Dinniman et al. (2012) <https://doi.org/10.1175/JCLI-D-11-00307.1> , Wåhlin et al. (2010) <https://doi.org/10.1175/2010JPO4431.1>; Schmidt et al. (2023)

<https://doi.org/10.1038/s41586-022-05691-0>; Thoma et al. (2008)

<https://doi.org/10.1029/2008GL034939>

L66 – I recommend using the full name ‘Berry Glacier’ here, since this is its first mention in the main text.

L67 – ‘locus’ suggests a single point; consider using ‘area/region’

L75 – The dash between 0.6 and km can be removed here, and similarly in other places where it is not used as an adjective.

L79 – It is not clear where this observation is shown on Figure 2; I suggest adding lines/labels

L87-98 – This analysis would be strengthened by discussing potential reasons for the lack of correlation between tidal forcing and short-term GL migration, as well as considering other controls that may drive this short-term variability. Comparing to studies that have documented different patterns of short-term GL migration elsewhere in Antarctica (e.g. Rignot et al., 2024; Chen et al., 2023; Freer et al., 2023) would be valuable. For example, could this interpretation be affected by calculating modelled tides 50-100 km seaward of the IGZ, where tidal forcing may be out of phase with the resulting tidal flexure within the IGZ itself? Exploring whether there any correlations with the longer-period spring-neap tide cycle, phase lags between rising and falling tides, or interactions with upstream subglacial hydrology – as observed in the Thwaites IGZ (e.g. Chartrand et al., 2024; Rignot et al., 2024) – would enhance the analysis. I note that Figure 5b shows several subglacial channels upstream of the IGZ, but this has not been addressed in the text.

L87-90 – It would also be interesting to see a comparison of the modelled Δ tide (at this same position 50-100 km seaward of the GL) compared to the measured InSAR Δ SSH *within the IGZ*. This may provide some insight into the non-linear relationship between tidal forcing further from the ice front and the ice shelf flexural response/ GL migration within the IGZ.

L97 – I suggest it is challenging to confidently state that the GL was "predominantly" in State 1 in 1996 based on only two observations. Perhaps rephrasing to reflect this uncertainty would be provide more context.

L101-115 – This section is somewhat challenging to follow as currently structured. Consider re-organizing and re-wording to improve clarity. It appears to mix different time periods (2006-2022 and 1996-2022), various regions (e.g., "between states 1 and 2," "west flank," "between the 1996 and 2008 GLs," "from the 2008 GL to 28 km beyond the 2021 IGZ upper limit," "along profiles G-H and C-D," "glacier on average"), and different values (ice elevation, thickness, thinning, speed, and speed-up). This makes the text difficult to follow. Additionally, when ∇h , ∇H and ∇V are averaged over specific areas, these areas should be clearly defined or located on a map.

L104 – Clarification is needed regarding how glacier speed changes were calculated; not enough detail is given here or in the methods section. Are these glacier speed changes calculated as an

average along flowlines, at specific points, or averaged across specific areas? Including this information in the text or figure would provide useful context.

L105 – The sentence on glacier *speed* feels out of place between other sentences discussing glacier *speed-up*. Consider moving it to a more logical position or removing altogether. It is important here to be specific about the time stamp of the velocity value.

L106 – The 1996 and 2008 GLs are not currently shown on Supplementary Figure 1, but should be added.

L107-109 – It is important to clearly define the region over which the average speed-up of 134 m/yr and a 59 m elevation decrease is calculated.

L110-111 – The results in L100-109 discussed change in V and h over the period 2006-2022, but it now seems for the flowline results you switch here a longer period 1996-2022. Please explain this discrepancy in time periods, and ensure the text explicitly mentions these timeframes to avoid confusion.

L110 – Consider rephrasing for clarity: ‘... the surface elevation decreased *by* up to 59 m and ice sped up *by* 48% from 448 m/yr in [YEAR] to 665 m/yr in [YEAR]’

L113 – is ‘on average’ referring to the entire glacier basin?

L113 – As explained in the general comments above, I suspect these thinning rates are significantly overestimated given the region is not in hydrostatic equilibrium (especially if these values are referring to the entire glacier basin, the majority of which is fully grounded).

L114-15 – Is this referring to the *spatial* pattern of thinning across the whole basin? I suggest you are more explicit about what is being cited here from Selley et al. (2021); is this an observation they also made?

L116-123 – I recommend the authors clarify the time period and locations for ice discharge calculations, and how this relates to the two time periods of observed h and V change discussed above (2006-2022 vs 1996-2022). It is currently difficult to ascertain how this was done.

L116 – Against which baseline are these discharge anomalies measured? Is it the same 1979-2004 reference period cited for the SMB anomalies in the following sentence? If so, please clarify.

L119 – Consider rewording for clarity: ‘Between 1996 and 2022, annual anomalies in mass balance, M , increased by a factor of 6.6, from X Gt/yr to Y Gt/yr’

L122 – It would be more logical to introduce the total increase in ice discharge at the start of this paragraph, and then transition from discharge > SMB > mass balance.

L127 – ‘jack up’ is a somewhat colloquial phrase which can have negative connotations; a different term like ‘lift up’ would be more suitable here.

L131 – The interchangeable use of "OIB" and "MCoRDS" in the text can be confusing. Consider specifying "MCoRDS" when referring to radar-derived thickness data and "ATM" for altimetry data from OIB and updating this consistently throughout.

L141 – It would help to clarify here that you are comparing results to a modeling study, not direct observations. Consider rewording to something like: "... gravity gradients, as *modeled* in [20]."

L144 – A number of previous studies have also observed this; you may wish to cite works such as Chen et al. (2023), Freer et al (2023), Rignot et al. (2024).

L146 – Please specify the time period over which basal melt rates were calculated – was this solely for 2019-2022, as indicated in Figure 5?

L154 – It would be helpful to clarify which area is being referred to here and to mark this region on Figure 5.

L156 – you should use “meters” instead of “m’s”

L171-173 - There appears to be a contradiction between these sentences. The text initially states that the Berry IGZ was unknown during previous studies, but then references comparisons of melt rates "in the IGZ of Berry" between 2003-2008 and 2019-2022. Please clarify this.

L172 - The “area of coincident measurements” for comparing the three time periods of basal melt rates is unclear. I suggest the authors label this area in Supplementary Figure 5. This applies to the areas used for the melt rate calculations at the other glaciers, such as De Vicq, as well.

L176-177 – The statement “*the melt rates in the IGZ of Berry are also the largest for the Getz Ice Shelf*” appears in a section discussing melt rates in the 10 km downstream of the IGZ, rather than the IGZ itself. Consider moving this sentence to the previous section or clarifying that it refers to the downstream zone.

L177-180 – I suggest adding a reference to Supp. Fig 5d here, as that is the only place this result is shown

L182 – There should be a unit given in ‘... from 0 m at state 3 to...’

L185 – Does the 0.6 m value represent a calculated average of the observed InSAR displacements? Or is this a prediction? More clarity (+associated uncertainties) is needed.

L186 – the stated water flux of $6.9 \times 10^7 \text{ m}^3$ in 6 hours would assume that the GL migrates all the way to state 3 each tide cycle. However, your results indicate there is no observed correlation

between tide forcing and GL migration distance (Fig 2) – so do you think this is still a reliable estimate of water flux? Please clarify and/or be more open about the associated limitations of this assumption.

L186 – You use an area of 114 km² for the IGZ here, but earlier a 100 km² IGZ area was used to calculate melt rates (L154)? Clarifying why these values differ would be helpful.

L187 – This is the first and only time you mention a 2018 CTD, but there is not associated citation. Earlier, you referenced a 0.7C ocean temperature at 1,000m depth in front of Berry measured from a 2007 CTD (Jacobs et al. 2013). The '2018 CTD' data gives ocean thermal forcing at 1,000m depth of +3.6C. Is this a mistake in the text with the date, or are you referencing two different datasets here? If this is the case, please clarify the date and location, alongside the appropriate citation.

L188-189 – I suggest more detail is needed regarding the calculation of ocean heat and freshwater fluxes, including any associated uncertainties or assumptions.

L190 – the statement that 'only 30% of the heat will be used to melt ice' is based on modelling at Petermann Glacier in Greenland. How confident are you that this can also be applied in this setting? Without providing uncertainty estimates or being open about these limitations, this quantification could be misleading.

L191 – Please clarify the basis for the statement that the seawater intrusions may be "thicker by a factor of two"; has this been quantified or just speculative?

L206 – In general, the methods lack sufficient detail to reproduce the analysis, although perhaps this has been limited by the word count of the article. Citations should be provided for all datasets.

L218 – I suggest this should cite a paper describing the Sentinel-1 instrument, rather than an ice sheet study that has also used Sentinel-1.

L222-228 – This section is missing a description of how the differential SSH was calculated from the interferograms, and whether the GLs were delineated manually or using an automated approach. Also, how do you define a 'high quality' interferogram?

L226 – should be 'dense' instead of 'densely'

L226 - The attached data spreadsheet includes a column for GL migration distance ('GL migration_Berry') as well as columns of 'positive error' and 'negative error', however there is no description of these errors in the text. I suggest the authors include a robust description of these error terms and the calculations used to derive them here.

L249-258 – The temporal coverage of each of the elevation change products used here should be provided:

- ICESat-2 ATL14 DEM is timestamped to Jan 1 2020
- ICESat-2 ATL15 elevation change products – between 2019 and 2024

- Add date ranges for other satellite missions used in the MEaSUREs elevation change products

Please clarify whether you use the ATL15 ICESat-2 elevation products (1km) as well as the MEaSUREs product using ICESat-2 (2 km)? Or just the former? And this section is missing information and citation for the WorldView-derived DEM and ATM data (e.g. used in Fig. 3, and in calculation of ice bottom and bed topography below).

L262-3 – this seems to be a new result that hasn't been presented in the main text? Please clarify.

L278 – The current description lacks sufficient detail. In the main text it says the bed topography is reconstructed *'based on the minimum depth of ice that ensures flotation of ice over the zone of flexing, i.e. where a meter of change in SSH is sufficient to jack up the ice surface'*. Please elaborate here on this calculation, specifying e.g. the geoid used and whether the reconstruction was performed over the whole IGZ region, or along individual flowlines?

L281 - Given that FAC values are known to be uncertain over GZs, it would be important to acknowledge this uncertainty here (even if it cannot be precisely quantified). Additionally, consider citing the firm model used in BM3.7 to provide more context.

L283-286 – see previous comment on this interpretation; I would suggest it's more a case that your calculations are inconsistent with the radar measurements (rather than the other way round). A stronger justification is required, beyond simply that the L2 data is 'not as robust'.

L289 – It would be useful to see a justification of applying a Eulerian vs Lagrangian approach here, or acknowledgement of the associated uncertainties when applied to floating ice.

L293-4 – Please provide more detail on this, as per previous comment.

L295-299 - It is good that the potential overestimation of melt rates is acknowledged here. However, this point should be emphasized more strongly within the main text, accompanied by a more detailed explanation of the uncertainty estimation process. Specifically, it is unclear how you arrived at the stated uncertainty of "about 36%". It would be helpful to clarify this methodology, and any reported uncertainty values should be presented alongside the corresponding results in the main text for transparency and clarity.

Figures

In general, the figures and captions (in main text and supplementary) deserve some attention to bring up to publication standard. Most text is too small to read comfortably, there are a mix of font types and sizes, and overlapping data and sub-panels obscures some important information. I recommend using a consistent structure for figure captions throughout, i.e. Fig X - main title. (a) subpanel caption; (b) subpanel caption; (c) etc. I have made some suggestions here for a number of (relatively minor) edits that could be made to the main text figures that I believe would significantly improve their quality. The same principles should be applied to the supplementary figures.

Figure 1:

This figure contains a lot of valuable and important data, but its impact is currently limited by the way it is presented. I suggest that the authors should consider re-formatting to a portrait layout, which would allow it to be displayed as a full-page figure. Alternatively, consider separating panel (a) as its own figure, and organizing panels (b-g) into a separate portrait figure (e.g. a 2x3 grid). Some suggested improvements include:

- *Panel (a)* –
 - Avoid use of <> symbols in the velocity legend for clarity.
 - The colors of the flux gates get a bit lost; perhaps keeping these in bold black while reducing the opacity of the rest of the OIB lines would improve clarity.
 - Add a box to the map to show the area in panels (b-d).
 - The elevation contour values could be removed; they are somewhat redundant alongside the blue elevation color scale. This would simplify things and avoid contour values overlapping the main data you are presenting (i.e. the GL positions).
- *Panels (b-d)*
 - These panels are well-presented, but the caption should provide more information (i.e. explain these are interferograms composed from images sampled from the 3-4 tidal states shown in panels (e-g); and describe what the color bar represents).
 - Adding labels for the maximum tide height and/or differential SSH of each interferogram could assist readers in understanding the data presented.
 - Clarify whether the inclusion of three interferograms comprised of scenes all sampled at low tides is a deliberate choice; if so, consider briefly explaining this in the caption as it is interesting for the interpretation!
- *Panels (e-g)*
 - Consider reducing the number of values on each axis label (e.g. intervals of 12h on x-axis) to allow for an increase in text size.
 - The grey lines pointing to sections of the longer tide time series clutter the figure. Consider removing or ensuring they do not cross over one other.
 - Consider repositioning the inset panels to prevent them from covering important data on the main panel, as this is an issue across all three panels (e-g) where some of the inset panels cover the red dots.

Figure 2

- *Panel (a)*
 - This density plot is a very effective way to illustrate the three GL states.
 - Some clarification is needed for the values presented in the DInSAR/HE box. Consider adding details to the caption, including an explanation of HE as hydrostatic equilibrium, and incorporating lines on the map to show the locations of the HE GLs for better reference.
- *Panel (b)*
 - Is there a particular reason for using a different color map and phase difference scale compared to Figure 1(b-d)? I'd say the color scale in Figure 1(b-d) is clearer.

- From your caption, I would expect the downstream seawater intrusions to be marked by a region of uplift, and upstream extrusion of seawater trapped in a prior cycle marked by a region of subsidence – but it appears that these labels are reversed on the map? Or is the region of extrusion further upstream (between states 2 and 3)? This may be my own misinterpretation, but it would be good to clarify this. Adding labels to show the regions of intrusion vs extrusion would be beneficial.
- Including arrows above/below the legend to illustrate the meaning of the diverging fringe colors could greatly enhance comprehension of this figure for those less familiar with DInSAR maps (i.e. pink>yellow>blue = uplift; and blue>yellow>pink = subsidence).
- *Panel (d)*
 - This is an effective presentation of these results. A few minor clarifications would further aid interpretation:
 - Indicate what the error bars represent (both for modelled max tide and GL migration distance).
 - Confirm that h_{\max} refers to the maximum modelled tide and clarify in the caption.

Figure 3

- The caption currently lacks clarity, specifically,
 - L2 – Clarify that (a) and (b) represent *change* in h / *change* in V
 - L3-4 - Remove 'Year 2006 is the reference for (a-b)'; this info is already conveyed in the preceding text and figure.
 - L4-5 – Maintain consistent phrasing for panels (c,d) and (e,f), i.e. ' $V(h)$ along profiles G-H and C-D *for each year* 1996-2022'
 - L6 – specify that the elevation data is from the WorldView (WV) DEM (i.e. not just from WV imagery)
 - L6-12 – separate captions for panels (g) and (h); currently confusing as different datasets are used in each for the same thing.
 - L6-12 – Clarify descriptions of the different blue lines ('blue line'/'blue dash line'/'blue solid line'/'dark blue line'). It is confusing where the same line style is used in panels (g) and (h) to show different data (e.g. in (g) the solid light blue line is from the 2019 WV DEM, but in (h) the same line style shows the '2012 DEM'). Consider changing the line styles or providing a more explicit legend.
 - L7 – please clarify which DEM is used for ice draft in (h)? It is unclear what instrument is used for the '2012 DEM' – it says ICESat-2 in the caption, but this cannot be correct as ICESat-2 only launched in 2018.
 - L8 – please clarify if by 'flotation from ICESat-2' you mean: *flotation depth derived from the 2020 ICESat-2 DEM?*
 - L10-12 – ensure consistency in the description of the OIB data points
- *GL positions/IGZ zones*

- On panels c, e and g, it is confusing having both the horizontal and vertical bars showing GL locations, positioned both above and below the main data. I suggest keeping just the horizontal bars for 2019-21 to show the short-term range of GL positions in each of the 'states' and labelling the three states in each panel (as has been done in g). Then you can use a vertical bar/dot for 1996 and 2008/9 GL positions where you don't have sampling of the IGZ length.
- For panels d, f and h, it is fine to use vertical lines, as this is showing the total width of the glacier/shelf.
- I suggest positioning all GL positions at the bottom of each panel (including in g) for consistency, rather than having some above and below the main data.
- Reflect any changes made in the caption
- *Panels g-h* – These are currently quite confusing to interpret.
 - Consider moving both the x-axis and IGZ labels to the bottom of each figure (rather X than in the middle of the glacier).
 - Clarify if elevation measurements are relative to mean sea level
 - Legend and caption currently missing info on the ?s, striped areas and blue shaded areas.
 - It is currently unclear what the areas labelled as ice/bed/water are using data from? BedMachine? Or the bed you've deduced from flotation? In (h) looks like the 'bed' is from the OIB data, but in (g) it's from a mixture of BedMachine and your WV-DEM-derived bed.
 - In panel (g), the blue dots representing the OIB data are not easily visible. The legend entry for OIB reads like a date (i.e. 2011/12/16 could be interpreted as 16 December 2011). I suggest since there are only four OIB points on this panel, each could be individually labelled with their measurement year.
 - There doesn't seem to be a consistency in the results at the expected crossover point between profiles G-H in panel (g) and C-D in panel (h); perhaps this is the result of using the same colors in both panels for different data?

Figure 5

- The sizing of this figure is much better
- On panel (b) red GLs get lost in the red melt areas; consider using a different color scale?

Datasets

The authors have provided easily accessible datasets of derived GL positions, flux gates etc. that were straightforward to open in GIS software. I recommend that all processed interferograms should also be included in the dataset. The excel spreadsheet '*GL analysis, tidal heights, speed & surface height.xlsx*' contains a lot of useful data, but some clarification and reorganization is required to improve its usability. Notably:

- Columns F and G ('positive error' and 'negative error') appear to indicate the number of days between the first and last SAR image date and the midpoint between these dates. Please clarify how this constitutes an error, and—if so—how this error is dealt with in the subsequent processing.
- Columns P and Q are also labelled 'positive error' and 'negative error', but these appear to indicate some error in the derived GL position. Please provide an explanation of this error and its calculation, assumptions etc.
- This dataset contains columns for modelled tides/IBE at 50 km and 100 km. Please clarify what these distances refer to – i.e. is it the distance between the downstream IGZ limit and the position used to calculate the tide? If so, which of these locations (50 or 100 km) did you use? If only one location is used for the tide calculation, only these values should be included in the dataset to avoid confusion.
- It is confusing that the SAR image acquisition dates are repeated three times in the same Excel sheet: in columns A-D, S-V and AP-AS. There is also repeated GL migration data in columns O ('GL migration_Berry') and AM/AN ('GL migration_Sentinel-1'/'GL migration_COSMO-SkyMed'). Consider consolidating these columns to remove unnecessary duplication.

Code availability

- The Zenodo link points to code used to compute basal melt rates at Thwaites, originally from a different paper (Gadi, 2024). I am not sure of the specific journal guidelines on this, but I suggest this should either be appropriately cited alongside the dataset, or specific code used for this work at Berry should be provided.
- No code is provided for the other calculations described in the methods, including ice bottom/bed topography, ice mass balance, GZ length or tides. This should be included to ensure full transparency.

Second round of review

Chen et al. - Rapid retreat of Berry Glacier, West Antarctica, linked to seawater intrusions revealed by satellite radar interferometry data

Reviewer 1 - Response to Response to Review

General comments

I commend the authors for their thorough revision of this manuscript, and I agree that it is now significantly improved! The title is much improved and the majority of my concerns have been addressed, regarding the text, methods descriptions, and the treatment of uncertainties. The figures are in much better shape too. I have one major remaining concern regarding the discussion about the cavity formation (detailed below), and a few additional minor comments / suggestions that the authors may like to consider before final submission. Once addressed, I would recommend this manuscript is acceptable for publication.

I commend the authors for their thorough revision of this manuscript – it is now substantially improved. The revised title is clearer, and the authors have addressed the majority of my previous concerns regarding the text, methodological descriptions, and treatment of uncertainties. The figures are also much improved. I have one remaining major concern related to the discussion of cavity formation (detailed below), along with a few additional minor comments and suggestions for the authors to consider prior to final submission. Provided these points are addressed, I would recommend the manuscript for publication. Congratulations.

Major Comments

Discussion of the Berry Cavity Formation

I appreciate the additional explanation made in the response document regarding the interpreted formation of a >100m deep cavity in the Berry IGZ, however I don't think this has been sufficiently reflected in the revised manuscript. In your response, you state:

“the cavity has areas of partial grounding, and the water column must be thicker in between these pinning points, perhaps 10’s m, but we do not have data to quantify it”.

However, this doesn’t seem to match up with what is written in the manuscript; i.e.,

L172 ‘During the past 25 years, a cavity several 100 meters in size has formed beneath Berry, up to 300 m near state 2’;

L205 ‘From the modeled tidal height of ± 1.2 m (Fig. 2d), we estimate that the average water in the IGZ cavity must be at least 0.6 m”

L226 “Most likely, the thickness of the water column coming in and out of the cavity must be greater than 0.6 m, possibly by a factor of 2.5”.

Perhaps this discrepancy is related to your comment made in the response document:

“The DInSAR fringe patterns observed in this region reflect the vertical displacement of the ice surface driven by subglacial water motion, rather than the total water thickness”.

I would argue that this distinction is still not made clear enough in the manuscript: Is the 0.6 m value relating to the *change* in depth in the cavity as measured by the DInSAR, on top of an absolute cavity depth of >100m? And how does this relate to the pinning points? To address this before publication, the authors should:

1. Reconcile these estimates for cavity depth, both the distinction between the 0.6m and >100m values, as well as whether the cavity depth should be on the order of 10s or 100s of meters.
2. State that you don’t have the data to accurately quantify the depth of the cavity (as you say in your response).
3. Soften the language used to describe your interpretations of the cavity formation (i.e. make it clear that the *observations suggest* that a cavity >Xm thick has formed at state 2, rather than stating it as definitive fact.).

Minor Comments

Use of HE for calculating thickness and basal melt

Thank you for explaining your application of hydrostatic equilibrium for ice thickness and basal melt estimates. In particular, the following response was very helpful for understanding and justifying your approach, and should therefore be included in the methods section of the paper. *“We apply the hydrostatic equilibrium (HE) assumption only to regions where seawater intrusions are inferred at a specific time period. For the period 1996–2005, changes in ice thickness were equated directly with surface elevation changes, as the GL reached to state 2 around 2006. Between 2006 and 2011, the HE assumption was applied within the region from state 1 to state 2. From 2012 to 2022, HE was applied across the entire 2021 IGZ extent.”*

Specific comments

L205-225 - I appreciate the edits to this section speculating on max melt rates - it is much stronger now. In your response to my initial comments, you state: *“This calculation is a maximum melt rate if water intrusions flush the entire cavity and all available heat is used to melt the ice”*. This is a critical statement that should be included in the text here - either at the very start of this paragraph or in the last few sentences.

L226-240 (broader implications) - This new text is a valuable addition to the paper. I have a couple of minor suggestions for clarity:

- I suggest rewording the first sentence to something like : *‘While the relatively small ice volume of Berry Glacier limits its potential sea level rise contribution, our results have important implications for ice sheet modeling in Antarctica’*
- L233 - I am not clear on what *‘thinning the strand’* means. Please clarify.

L376 - I suggest you should add the additional information about the 4.5 km spatial filter that you provided in your response to the manuscript text at L376-377 (*‘The window size is 4.5 km, which roughly corresponds to the size of the bending zone discussed by Chartran et al. 2024’*)

Figure comments

Fig 3

- Panels G and H are much clearer and easier to interpret now! A couple of minor remaining points:
 - Still missing legend showing what stripey blue areas is
 - I presumed that the shaded blue area was error bounds on the IS2 and WV-derived ice bottom elevations, but perhaps I am mistaken. In the methods (L359) you write *“The blue shaded areas in Fig. 3g mark the regions where vertical motion from seawater intrusion is detected, but the bed depth is unknown”* – this is a very helpful description that is important for the interpretation of the figure, so I suggest you include a similar sentence in the figure caption to clarify this for readers.

Fig 5

The addition of white boxes showing the “area of coincident measurements” is very helpful here! Please make sure this is sufficiently described in the caption though.

Supp. Fig. 7

- The additional information about tides on each panel is very useful.
- I suggest labeling each panel with the date e.g. 1996a, 1996b, 2008a etc, so this ties in more directly with the table below.